# ENHANCING PRE-TRAINED REPRESENTATION CLASSIFIABILITY CAN BOOST ITS INTERPRETABILITY

**Shufan Shen**[1,2]**, Zhaobo Qi**[3]**, Junshu Sun**[1,2]**, Qingming Huang**[1,2,4]**, Qi Tian**[5]**, Shihui Wang**[1,4*]

[1] Key Lab of Intell. Info. Process., Inst. of Comput. Tech., CAS
[2] University of Chinese Academy of Sciences  [3] Harbin Institute of Technology, Weihai
[4] Peng Cheng Laboratory  [5] Huawei Inc.
{shenshufan22z, sunjunshu21s, wangshuhui}@ict.ac.cn
qizb@hit.edu.cn    qmhuang@ucas.ac.cn    tian.qi1@huawei.com

## ABSTRACT

The visual representation of a pre-trained model prioritizes the classifiability on downstream tasks, while the widespread applications for pre-trained visual models have posed new requirements for representation interpretability. However, it remains unclear whether the pre-trained representations can achieve high interpretability and classifiability simultaneously. To answer this question, we quantify the representation interpretability by leveraging its correlation with the ratio of interpretable semantics within the representations. Given the pre-trained representations, only the interpretable semantics can be captured by interpretations, whereas the uninterpretable part leads to information loss. Based on this fact, we propose the Inherent Interpretability Score (IIS) that evaluates the information loss, measures the ratio of interpretable semantics, and quantifies the representation interpretability. In the evaluation of the representation interpretability with different classifiability, we surprisingly discover that *the interpretability and classifiability are positively correlated*, *i.e.*, representations with higher classifiability provide more interpretable semantics that can be captured in the interpretations. This observation further supports two benefits to the pre-trained representations. First, the classifiability of representations can be further improved by fine-tuning with interpretability maximization. Second, with the classifiability improvement for the representations, we obtain predictions based on their interpretations with less accuracy degradation. The discovered positive correlation and corresponding applications show that practitioners can unify the improvements in interpretability and classifiability for pre-trained vision models. Codes are available at here.

## 1 INTRODUCTION

With the rapid development of vision pre-training models, their representations have been employed in diverse scenarios (Li et al., 2022a; Kirillov et al., 2023; Cui et al., 2024), achieving remarkable performance on downstream tasks (Dosovitskiy et al., 2021; Liu et al., 2021; Tong et al., 2022; Wang et al., 2023). However, except for better classifiability, the extensive applications that require reliable predictions underscore the interpretability of vision representations. To this end, interpretability-oriented methods (Koh et al., 2020; Wang et al., 2021; Chen et al., 2024) impose pre-defined semantics on each dimension of the representation space. These semantics serve as the interpretability constraints for further predictions, different from the training strategies that solely prioritize classifiability improvement (Dosovitskiy et al., 2021). However, the pre-defined semantics gain representations with interpretability but degraded classifiability, indicating an *inherent conflict between interpretability and classifiability* (Rudin et al., 2022; Zarlenga et al., 2022).

Different from the interpretability-oriented representations, the other type of methods (Kim et al., 2018; Ghorbani et al., 2019; Oikarinen et al., 2023) obtain interpretations by capturing interpretable semantics within the classifiability-oriented pre-trained representations. However, in contrast to the pre-defined semantics for the interpretability-oriented representations, the semantics of the

---

[*]Corresponding author.

classifiability-oriented representations are not guaranteed to be interpretable and fully captured during interpretation. As a result, semantic information loss occurs when only a subset of the original semantics can be interpreted (Yüksekgönül et al., 2023). The interpretability difference between the classifiability-oriented representations facilitates us to further explore the interpretability-classifiability conflict: first, whether the reduced interpretability of the classifiability-oriented representations is due to their high classifiability; second, whether the improvements in the interpretability always lead to reduced classifiability. Our target is to answer that *can the classifiability-oriented representations achieve high interpretability and classifiability simultaneously?*

In this paper, we present studies on the aforementioned question and discover that *interpretability and classifiability are not conflicting but positively correlated for classifiability-oriented representations.* To quantify the representation interpretability, we introduce the Inherent Interpretability Score (IIS). IIS is defined as the ability of representations to preserve task-relevant semantics (*e.g.*, concepts and patterns useful for classification) in interpretations (Figure 1). More interpretable representations should have less semantic information loss in interpretations. For the representation classifiability, we employ prediction accuracy as the evaluation metric. With accuracy and our proposed IIS, we investigate the interpretability-classifiability re-

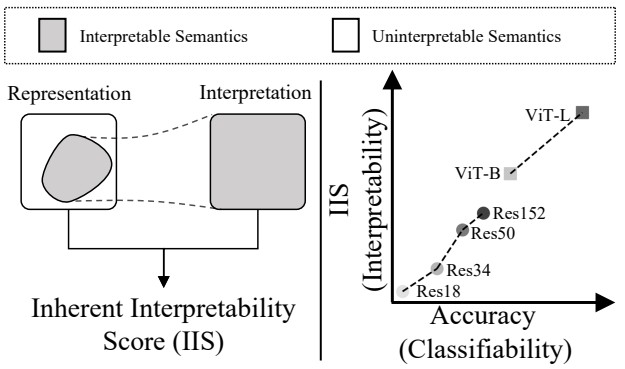

Figure 1: Classifiability-oriented representations have uninterpretable semantics, causing an interpretability reduction. We propose the IIS to quantify representation interpretability with the maintenance of task-relevant semantics in interpretations *(left)*. By comparing the IIS and prediction accuracy of representations from different models, we observe a positive correlation between interpretability and classifiability *(right)*.

lationship of pre-trained representations. Extensive experiments on different datasets (Krizhevsky et al., 2009; Wah et al., 2011; Russakovsky et al., 2015) and pre-trained representations (He et al., 2016; Dosovitskiy et al., 2021; Liu et al., 2021; 2022) reveal a **positive** correlation between interpretability and classifiability, rather than a contradictive one as discussed in (Mori & Uchihira, 2019; Mahinpei et al., 2021; Zarlenga et al., 2022; Dombrowski et al., 2023).

The outcome of our study benefits two aspects. First, improving interpretability during representation fine-tuning can further promote the classifiability on downstream tasks. Second, representations with higher classifiability possess enhanced interpretability, which also indicates more interpretable task-relevant semantics. Therefore, predictions based on the interpretations become more comparable to the original classifiability-oriented representations and exceed that of the interpretability-oriented representations (Zarlenga et al., 2022; Yang et al., 2023). The mutual promoting relation between interpretability and classifiability provides new pathways for interpretable representation learning. The contributions of our work are as follows:

- We propose a quantitive measure (called IIS) for the representation interpretability.

- We uncover the positive correlation between the interpretability and classifiability of the classifiability-oriented pre-trained representations.

- We discover the mutual promoting relation between the interpretability and classifiability of the classifiability-oriented representations. Improving the interpretability can further improve the classifiability, and representations with higher classifiability are more interpretable to preserve more task-relevant semantics in interpretations.

## 2 REPRESENTATION INTERPRETABILITY MEASUREMENT

In this section, we propose IIS to quantify the representation interpretability, which measures the semantic information loss during interpretation by the accuracy retention rate. Given a dataset $\mathcal{D}$,

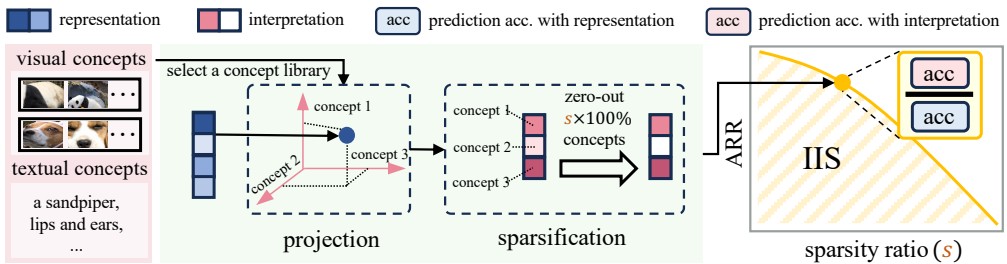

Figure 2: Definition and computation of the IIS. Given a pre-trained model and a downstream task, we first collect task-relevant concept libraries *(left)* and interpret model representations by projecting them into the concept space. By interpreting representations with sparse concepts, we can extract their interpretable semantics *(middle)*. The IIS is defined as the representation's ability to retain accuracy when predicting solely based on interpretations *(right)*.

a pre-trained vision model $f : \mathbb{R}^I \to \mathbb{R}^D$ that maps the inputs from $\mathbb{R}^I$ to representation space $\mathbb{R}^D$, we first interpret the representation space $\mathbb{R}^D$ by correlating it with $M$ human-understandable concepts which span a concept space in $\mathbb{R}^M$ (Section 2.1). Then we project the representation to $\mathbb{R}^M$ as interpretations, which can also serve for making predictions on downstream tasks (Section 2.2). IIS with the accuracy retention rate between the representation and interpretation predictions measures the semantic information loss from the representation space $\mathbb{R}^D$ to the constructed space $\mathbb{R}^M$, indicating the representation interpretability (Section 2.3).

## 2.1 BRIDGING REPRESENTATION SEMANTICS WITH CONCEPTS

The representation space can be interpreted by correlating to human-understandable concepts. To achieve this, we collect a task-relevant concept library with $M$ concepts inspired by previous methods (Kim et al., 2018; Ghorbani et al., 2019; Oikarinen et al., 2023). These concepts span a concept space $\mathbb{R}^M$, which can be employed for interpretations by correlating with the vectors in $\mathbb{R}^D$.

**Concept Library.** As the fundamental issue of interpretations, how to define the concept library has an important impact on our investigations regarding the interpretability-classifiability relationship. To ensure our investigations are not confined to a specific concept library, we perform investigations based on four types of concept libraries (*Prototype, Cluster, End2End, Text*) respectively. These libraries can be categorized into `Visual` and `Textual` based on the concept modalities. Note that different concept libraries are utilized separately during investigations without being concatenated.

`Visual`. A visual concept is represented by several image patches (Kim et al., 2018) or segments (Ghorbani et al., 2019). Here we take the patch as an example. Given a set of patches, a pre-trained vision model is utilized to extract visual concepts. This model takes each patch as input and generates a corresponding feature. Then the patches are aggregated into visual concepts based on their features. Specifically, we build three types of concept libraries according to their aggregation strategies. *(i) Prototype*: each patch serves as a concept without being aggregated. *(ii) Cluster*: patches are aggregated into concepts with pre-defined clustering methods applied to their features. *(iii) End2End*: patches are aggregated in a learnable manner during end-to-end training.

`Textual`. A textual concept is represented by a single word or phrase. The development of large language models (Brown et al., 2020) enables the automated collection of a *Text* concept library based on target classes. This capability stems from the domain knowledge encapsulated within these models regarding the salient concepts for detecting each class. In light of the success of previous work (Yang et al., 2023; Oikarinen et al., 2023), we utilize GPT-3 (Brown et al., 2020) to extract textual concepts for each class with pre-defined prompts.

**Projecting Relations between Representation Space and Concept Space.** Given a pre-trained vision model $f : \mathbb{R}^I \to \mathbb{R}^D$, an image classification dataset $\mathcal{D}$ and a concept library $\mathcal{C} = \{c_1, c_2, ..., c_M\}$ constructed based on $\mathcal{D}$, we can denote each concept $c_i$ with a vector $\mathbf{c}_i \in \mathbb{R}^D$. If $c_i$ is a visual concept comprised of $n$ image patches/segments $\{x_1, ..., x_n\}$, its concept vector $\mathbf{c}_i \in \mathbb{R}^D$ is obtained by averaging the representations of these patches/segments,

$$\mathbf{c}_i = \frac{1}{n} \sum_{j=1}^{n} f(x_j). \tag{1}$$

If $c_i$ is a textual concept, the corresponding vector $\mathbf{c}_i$ is obtained in a learnable manner. A vision-language model (Radford et al., 2021) $f_{\mathtt{vl}} : \mathbb{R}^I \times \mathcal{C} \to \mathbb{R}$ is utilized to generate the soft label $y_x^{c_i} = f_{\mathtt{vl}}(x, c_i) \in \mathbb{R}$ for concept $c_i$ on image $x \in \mathcal{D}$. The vector $\mathbf{c}_i$ is obtained through the minimization of the discrepancy between $y_x^{c_i}$ and the inner product of $\mathbf{c}_i$ and the image representation $f(x)$

$$\mathbf{c}_i = \arg\min_{\mathbf{c}_i^*} \mathbb{E}_x \left[ (y_x^{c_i} - f(x)^\top \mathbf{c}_i^*)^2 \right]. \tag{2}$$

The concatenated concept vectors $\mathbf{C} = [\mathbf{c}_1, \mathbf{c}_2, ..., \mathbf{c}_M] \in \mathbb{R}^{D \times M}$ can project representations from $\mathbb{R}^D$ to $\mathbb{R}^M$, empowering the interpretation of the representations in $\mathbb{R}^D$ with $M$ concepts.

## 2.2 Interpreting Representations with Sparse Concepts

With the concatenated concept vectors $\mathbf{C}$, the pre-trained representation $\mathbf{x} = f(x) \in \mathbb{R}^D$ of image $x$ can be projected into the constructed concept space $\mathbb{R}^M$,

$$\mathbf{x}^{\mathcal{C}} = g_{\mathcal{C}}(\mathbf{x}) = \mathbf{C}^\top \mathbf{x}, \tag{3}$$

where the $i$-th dimension of the projection result $\mathbf{x}^{\mathcal{C}} \in \mathbb{R}^M$ can be regarded as the contribution score of the concept $c_i$ to the original representation $\mathbf{x}$. However, these contribution values can not be employed as human-understandable interpretations due to the human cognitive limitations in processing a large amount of concepts (Ramaswamy et al., 2023). A concept library typically involves a large number of concepts to encompass task-related semantics comprehensively (Oikarinen et al., 2023; Yang et al., 2023). Therefore, incorporating the contribution of each dimension in $\mathbb{R}^M$ for interpretation would entail an overwhelming number of concepts, which counter-productively becomes incomprehensible for humans. To address this problem, we utilize a sparsification mechanism (Shen et al., 2024) to control the number of concepts that contribute to the interpretations.

Given a sparsity ratio $s \in [0, 1]$, the sparsification objective is to zero out $s \times 100\%$ elements in the projection result $\mathbf{x}^{\mathcal{C}} \in \mathbb{R}^M$. Based on Equation 3, the larger absolute values in $\mathbf{x}^{\mathcal{C}}$ indicate the greater contribution of the concepts and their stronger correlations with image semantics. Therefore, we zero out elements and sparsify the concepts based on the element values in $\mathbf{x}^{\mathcal{C}}$. Let $\mathbf{x}_{\tilde{s}}^{\mathcal{C}}$ denote the element with the $\lceil s \times M \rceil$-th smallest absolute value of $\mathbf{x}^{\mathcal{C}}$. For the $i$-th element $\mathbf{x}_i^{\mathcal{C}}$ in $\mathbf{x}^{\mathcal{C}}$, the sparsification can be formulated as,

$$\mathbf{x}_i^{\mathcal{C},s} = g_s(\mathbf{x}^{\mathcal{C}})_i = \mathbf{x}_i^{\mathcal{C}} \max \left( |\mathbf{x}_i^{\mathcal{C}}| - |\mathbf{x}_{\tilde{s}}^{\mathcal{C}}|, 0 \right), \tag{4}$$

where $\mathbf{x}^{\mathcal{C},s} \in \mathbb{R}^M$ denotes the interpretation of representation $\mathbf{x}$. This sparsification function removes concepts in ascending order based on their similarity to the image representations and ensures that only $\lceil (1 - s) \times M \rceil$ concepts serve the interpretation.

## 2.3 Inherent Interpretability Score

The semantics of the pre-trained representations are decoded with various concepts during interpretation, where more interpretable representations empower less semantic information loss (Sarkar et al., 2022; Yüksekgönül et al., 2023). Only the interpretable semantics are captured by the interpretations. As a result, interpretations with certain information loss become less discriminative and classifiable on downstream tasks compared to the original representations. In light of this, we formulate the evaluation of the representation interpretability as the accuracy comparison between predictions based on the original representations and their corresponding interpretations.

To evaluate the accuracy of predictions based on the interpretations, we further train a linear classifier $g_{\mathtt{cls}}$ that takes interpretations as inputs for prediction,

$$g_{\mathtt{cls}}(\mathbf{x}^{\mathcal{C},s}) = \mathbf{W}^\top \mathbf{x}^{\mathcal{C},s} + \mathbf{b}, \tag{5}$$

where $\mathbf{W} = [\mathbf{w}_1, \mathbf{w}_2, ..., \mathbf{w}_N] \in \mathbb{R}^{M \times N}$ and $\mathbf{b} \in \mathbb{R}^N$ are trainable parameters.

In comparison between the prediction accuracy of the original representations and their corresponding interpretations, we propose Accuracy Retention Rate (ARR), *i.e.*, the ratio of prediction accuracy based on interpretations to that on representations,

$$\mathrm{ARR}(f, g_{\mathtt{cls}} \circ g_s \circ g_{\mathcal{C}}, h, \mathcal{D}) = \frac{\mathrm{Acc}(f, g_{\mathtt{cls}} \circ g_s \circ g_{\mathcal{C}}, \mathcal{D})}{\mathrm{Acc}(f, h, \mathcal{D})}, \tag{6}$$

where $h$ denotes a linear classification head. $\texttt{Acc}(f, h, \mathcal{D})$ means the accuracy on dataset $\mathcal{D}$, achieved by a model with a pre-trained backbone $f : \mathbb{R}^I \rightarrow \mathbb{R}^D$ for representation extraction and a linear classification head $h : \mathbb{R}^D \rightarrow \mathbb{R}^N$ for classification.

Note that ARR is still not sufficient for the measurement of the representation interpretability. During the interpretation, we adjust the sparsity ratio $s$ through the proposed sparsity mechanism to control the amount of concepts to satisfy the rule of human-comprehensibility. However, determining the value of the sparsity ratio $s$ poses another challenge, where the optimal value depends on both the downstream task and subjective human judgment. To minimize these dependencies and ensure the broad applicability of the evaluation, we consider the values of $s \in [0, 1]$ and measure the total ARR across these values as the interpretability measurement. Formally, the interpretability metric IIS is the area under the curve of sparsity ratio $s$ and ARR,

$$\texttt{IIS}(f, \mathcal{C}, \mathcal{D}) = \int_s \texttt{ARR}(f, g_{\texttt{cls}} \circ g_s \circ g_{\mathcal{C}}, h, \mathcal{D})\mathbf{d}s. \tag{7}$$

In practice, given a pre-trained model $f$, a dataset $\mathcal{D}$ and a concept library $\mathcal{C}$, we select multiple sparsity ratios and train the corresponding $g_{\texttt{cls}}$ for each ratio to estimate IIS.

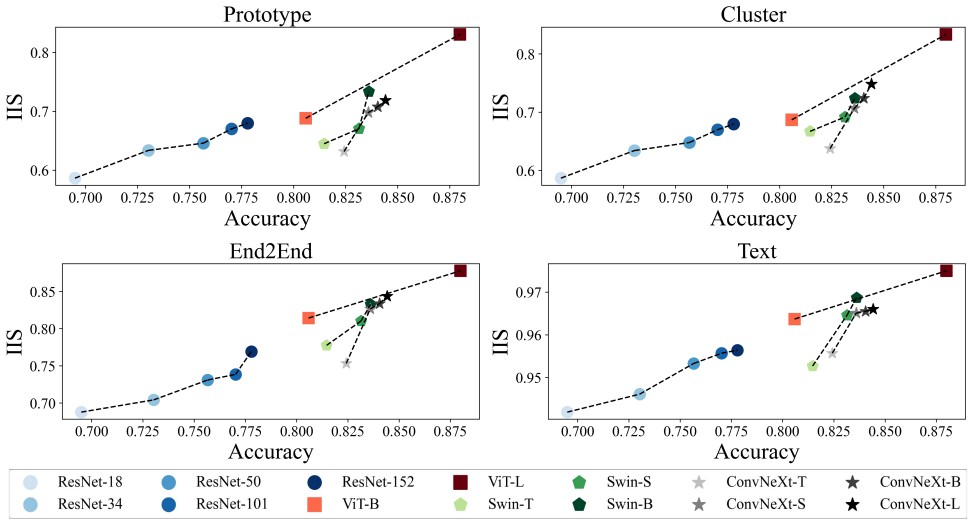

Figure 3: The relationship between IIS and the prediction accuracy of pre-trained representations on ImageNet. We provide experiments with four types of concept libraries.

# 3 INTERPRETABILITY-CLASSIFIABILITY RELATIONSHIP INVESTIGATION

First, we provide the experiment settings in Section 3.1. Then, we investigate the interpretability-classifiability relationship by comparing the IIS and prediction accuracy of pre-trained representations (Section 3.2). Finally, we analyze the relationships between classifiability and key factors of interpretability, namely, sparsity and ARR (Section 3.3).

## 3.1 EXPERIMENT SETTINGS

**Pre-trained Representations.** We quantify the representation interpretability of well-known pre-trained models including ResNet-18/34/50/101/152 (He et al., 2016), ViT-B/L-16 (Dosovitskiy et al., 2021), ConvNeXt-T/S/B/L (Liu et al., 2022), and Swin-T/S/B (Liu et al., 2021). The weights of these pre-trained models are provided by Torchvision (Paszke et al., 2019). In addition to image representations, we also conduct experiments on video representations (Carreira & Zisserman, 2017; Feichtenhofer et al., 2019; Tong et al., 2022; Wang et al., 2023; Li et al., 2023b;a) in Appendix A.4.

**Datasets.** We compute the IIS of representations on ImageNet1K (Russakovsky et al., 2015), CUB-200 (Wah et al., 2011), CIFAR-10 (Krizhevsky et al., 2009) and CIFAR-100 (Krizhevsky et al., 2009). These datasets cover diverse tasks. CIFAR-10/100 and ImageNet1K are widely used for

Figure 4: The relationship between the accuracy and IIS on three datasets (CUB-200, CIFAR-10, and CIFAR-100). The IIS is computed based on the textual concept library.

general image classification tasks. CUB-200 is specifically designed for fine-grained bird-species classification. Their sizes vary greatly, with CUB-200 having 5900 training samples, CIFAR datasets 50,000 each, and ImageNet1K having 1-2 million training images.

**Concept Library Construction.** For visual concepts, we aggregate image patches or segments into 200 concepts using three strategies illustrated in Section 2.1 on ImageNet1K. For textual concepts, we follow previous work (Oikarinen et al., 2023) that uses GPT-3 (Brown et al., 2020) to extract concepts with prompts. The number of concepts is as follows: 143 for CIFAR-10, 892 for CIFAR-100, 370 for CUB-200, and 4751 for ImageNet1K. We use CLIP (ViT-B-16) (Radford et al., 2021) to generate soft concept labels for all datasets.

More implementation details are presented in Appendix A.1

## 3.2 INTERPRETABILITY AND CLASSIFIABILITY ARE POSITIVELY CORRELATED

In this section, we explore the interpretability-classifiability relationship of pre-trained representations. Empirical analysis across representations from different models is first conducted, followed by concentrating on representations from the same model with different training iterations.

**Analysis across Representations from Different Pre-trained Models.** With representations of different pre-trained models, we investigate their interpretability-classifiability relationship with different concept libraries (Figure 3) and datasets (Figure 4). A positive correlation between classifiability and interpretability can be observed for all types of concept libraries and datasets, especially when comparing representations from models sharing the same architecture but different numbers of parameters (points linked by dotted lines, such as ViT-B and ViT-L). These results indicate that *the interpretability of pre-trained representations increases along with their classifiability improvement*. We observe the presence of this phenomenon in the pre-trained video representations as well (Appendix A.4). Additionally, we notice that the representation interpretability is also influenced by the model architecture, as evidenced by the non-strictly positive relationship between IIS and accuracy across representations from different architectures (*e.g.*, ViT-B and Swin-T). This observation reveals the potential of IIS in evaluating the interpretability of general model architectures.

**Analysis along the Pre-training Process.** To further investigate this positive correlation between interpretability and classifiability, we analyze the IIS evolution during the pre-training process of representations. Figure 5 depicts the changes in the IIS on different prediction accuracy and training epochs. In the initial phases of training, the representations achieve high IIS because both interpretations and representations have similarly low accuracy. This phenomenon does not affect the application of IIS in measuring the interpretability of pre-trained representations. However, as the number of training iterations increases, the IIS gradually increases with improved accuracy. Therefore, *the representation interpretability is primarily enhanced in the later stage of the training process*.

## 3.3 FACTOR ANALYSIS FOR INTERPRETABILITY

To provide an in-depth analysis of the interpretability, we decouple IIS into two key factors, *i.e.*, sparsity and ARR, investigating their mutual relationship and the relationship with classifiability.

First, we present the sparsity-ARR curve of pre-trained representations from different models on ImageNet in Figure 6, using four types of concept libraries. Considering representations from the same model, with the sparsity ratio $s$ increasing from 0 to 1, the interpretations rely on sparser concepts, and the corresponding values of ARR decrease. This indicates that sparser concepts lead to

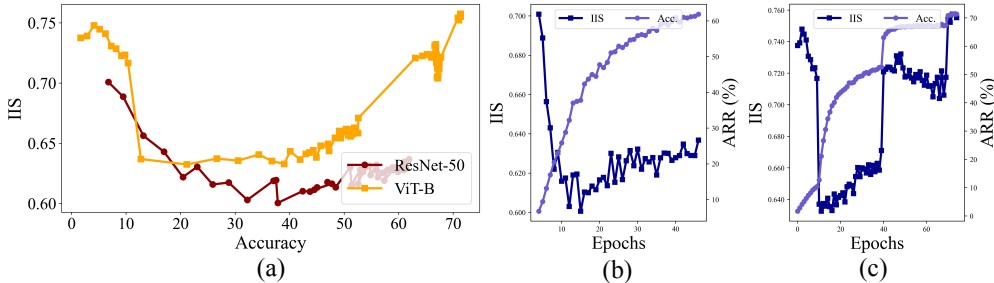

Figure 5: The evolution of the IIS with varying accuracy (a) and epochs during the pre-training process of representations from ResNet-50 (b) and ViT-B (c) on ImageNet.

more information loss in the interpretations of the same representation. Considering the comparison of different models, the representations show different abilities to support interpretations given different sparsity ratios. Representations that can better retain classifiablity (higher ARR) in interpretations under different sparsity ratios gain larger areas under the curve. This further validates IIS as the quantification metric of representation interpretability.

We further analyze the factor-level interpretability-classifiability relationship in Figure 7. ARR increases simultaneously with the improvement of the classifiability of the representation across different sparsity ratios. This indicates that *as the representation classifiability improves, it can be interpreted by sparser concepts with less task-relevant semantic loss (higher ARR)*. Based on these relationships, we can obtain completely interpretable predictions using concepts, with higher accuracy compared to interpretability-oriented methods. This can be further verified in Section 4.2.

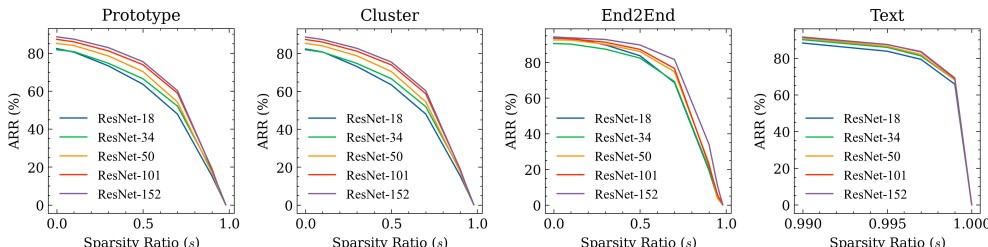

Figure 6: The sparsity-ARR curve of representations from multiple models with four types of concept libraries. All representations are evaluated on ImageNet.

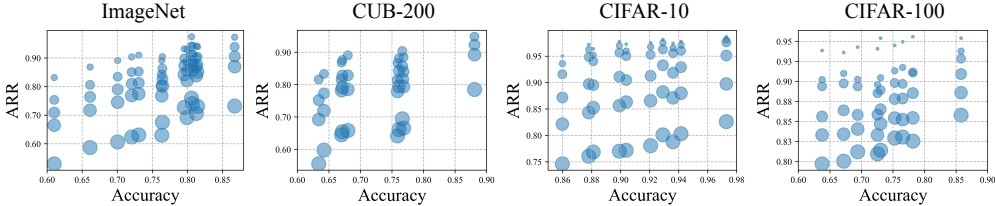

Figure 7: Relationship of the sparsity, accuracy, and ARR on different datasets. Larger point sizes indicate larger sparsity (*i.e.*, interpreting representations with sparser concepts). Experiments are conducted with text concept libraries specific to the downstream dataset.

# 4 APPLICATIONS OF INVESTIGATION RESULTS

The investigation results of Section 3 guide the application studies in two aspects. First, inspired by the positive correlation between interpretability and classifiability in Section 3.2, we can improve the classifiability of representations through interpretability maximization. Second, we offer interpretable predictions with higher accuracy based on the observation in Section 3.3 that improving representation classifiablity results in improved ARR across different sparsity ratios.

Table 1: The classification accuracy of widely-used pre-trained models and their accuracy after fine-tuning with IIS maximization.

| Model | Type | Acc@1 | Acc@5 |
|-------|------|-------|-------|
| ResNet-50 | origin | 75.66 | 92.67 |
|  | ours | **77.36** | **93.53** |
| Swin-T | origin | 81.46 | 95.78 |
|  | ours | **81.80** | **95.86** |
| ConvNext-T | origin | 82.42 | 96.18 |
|  | ours | **82.63** | **96.35** |
| ViT-B | origin | 80.58 | 95.15 |
|  | ours | **81.07** | **95.33** |

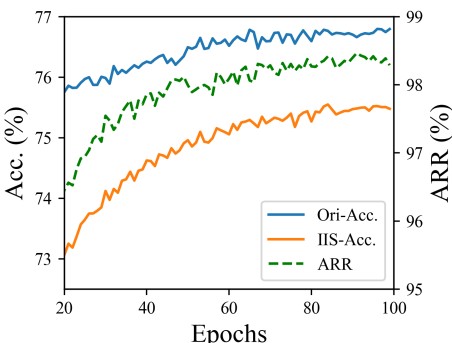

Figure 8: Prediction accuracy with representations in the original space (Ori) and a sparse subspace (IIS) during the fine-tuning of ResNet-50. The ARR is also provided.

Table 2: IIS maximization on ImageNet for 50 epochs. The sparsity ratio $s$ is set to 0.1.

| $M$ | 100 | **200** | 300 | 600 |
|-----|-----|---------|-----|-----|
| Acc@1 | 76.25 | **76.72** | 76.62 | 76.53 |
| Acc@5 | 93.08 | **93.20** | 93.12 | 93.16 |

Table 3: IIS maximization on ImageNet for 100 epochs. The dimension $M$ is set to 200.

| Sparsity Ratio ($s$) | 0 | **0.1** | 0.5 | 0.9 |
|----------------------|---|---------|-----|-----|
| Acc@1 | 76.62 | **77.07** | 76.95 | 76.67 |
| Acc@5 | 92.98 | **93.43** | 93.28 | 93.13 |

## 4.1 IMPROVING CLASSIFIABILITY BY INTERPRETABILITY MAXIMIZATION

Based on the interpretability (IIS) enhancement brought by classifiability improvement, this section, on the other hand, investigates whether enhancing IIS for representations would result in improved classifiability. Since the pre-trained model $f$ changes for every optimization step, computing IIS at each step can be time-consuming. To expedite the optimization process, we employ a simplified version of IIS, *i.e.*, replacing the concatenated concept vectors $\mathbf{C}$ with a learnable matrix $\mathbf{C}_l \in \mathbb{R}^{D \times M}$ and fixed sparsity ratio $s$. Maximizing the simplified IIS can improve the original IIS (Table A18), indicating that the simplified IIS can serve as a viable alternative to the original IIS during the optimization. With the simplified IIS, we fine-tune the model $f$ with the following objective:

$$\mathcal{L}(\mathbf{y}, (h \circ f)(x)) + \mathcal{L}(\mathbf{y}, (g_{\texttt{cls}} \circ g_s)(\mathbf{C}_l^\top f(x))), \tag{8}$$

where $\mathcal{L}$ denotes the cross-entropy loss function. The first term corresponds to the original prediction head, and the second term corresponds to the one with simplified IIS constraints. The simplified IIS is enhanced by maximizing accuracy with sparsity constraints in a learnable subspace. Table 1 presents the results on ImageNet. By incorporating the IIS maximization objective into the optimization, we observe a classifiability improvement for multiple widely used pre-trained representations. This underscores the mutual promoting relationship between interpretability and classifiability.

To better understand the classifiability improvement with IIS maximization, Figure 8 depicts the accuracy evolution for classification heads corresponding to the first and second terms in Equation 8 during the fine-tuning process. The evolution of ARR is also presented to analyze the interrelation between the original prediction head and the head with IIS constraints. With an increase in the number of training iterations, we observe a consistent improvement in the accuracy of both classification heads and their resulting ARR. This phenomenon can be explained by the regularization effect induced by the second terms in Equation 8, which encourages task-relevant semantics within $\mathbb{R}^D$ to be less complex and fit the low-dimension sparsified space in $\mathbb{R}^M$.

We further investigate the effect of hyperparameters in the simplified IIS. Ablation studies on the concept space dimension $M$ for $\mathbf{C}_l$ and the sparsity ratio $s$ are provided in Table 2 and Table 3, respectively. For the concept space dimension $M$, we observe that the model achieves optimal performance when $M$ is set to 200. For the sparsity ratio $s$, the optimal performance is achieved at a small setting ($s = 0.1$), while any higher ratios would result in performance degradation.

Table 4: Comparisons between interpretability-oriented methods and our methods on CUB-200. *We re-implement ECBM on our concept library.

| Method | #Concept | Acc@1 |
|---|---|---|
| Ante-Hoc | 312 | 64.17 |
| CBM | 112 | 75.90 |
| CEM | 112 | 79.60 |
| Proto2proto | 2000 | 79.89 |
| ProtopNet | 2000 | 80.80 |
| ECBM | 112 | 81.20 |
| ECBM* | 247 | 83.21 |
| Ours (ViT-L) | 123 | 81.30±0.08 |
| Ours (ViT-L) | 247 | **83.50**±0.15 |

Table 5: Accuracy of interpretable predictions (Acc_Inte), accuracy of pre-trained representations (Acc_Ori) and ARR on ImageNet.

| Method | Acc_Inte | Acc_Ori | ARR |
|---|---|---|---|
| SENN | 58.55 | - | - |
| Ante-Hoc | 65.09 | - | - |
| LaBo | 83.97 | - | - |
| Ours (Swin-T) | 78.72 | 81.46 | 96.63 |
| Ours (Swin-S) | 81.09 | 83.14 | 97.53 |
| Ours (Swin-B) | 81.88 | 83.61 | 97.93 |
| Ours (ViT-B) | 78.50 | 80.58 | 97.42 |
| Ours (ViT-L) | **86.85** | **87.99** | **98.70** |

## 4.2 PROVIDING INTERPRETABLE PREDICTIONS WITH HIGH ACCURACY

Section 3.3 shows that representations with higher accuracy can be interpreted by sparser concepts with less semantic loss. In light of this, we offer interpretable predictions based on $g_{\texttt{cls}}$ in Equation 5 with higher accuracy that is close to the original representations. Table 4 presents comparisons between predictions based on the interpretations of classifiability-oriented representations and the interpretability-oriented methods that train representations with interpretability constraints. Our interpretations achieve comparable or even better accuracy to interpretability-oriented methods (81.29% *vs.* 81.20%) with a similar number of concepts $M$ (123 *vs.* 112). As $M$ increases from 123 to 247, we observe a significant accuracy improvement over the existing best result (83.50% *vs.* 83.21%). Note that we only take the classification head as the trainable module, in contrast to interpretability-oriented methods training the entire model. This demonstrates the advantage of low training costs using classifiability-oriented representations to produce interpretable predictions.

Furthermore, we report the accuracy comparison on ImageNet in Table 5. Compared with existing interpretability-oriented methods, the interpretations of classifiability-oriented representations surpass them by 2.88% (86.85% *vs.* 83.97%). Moreover, we compare the accuracy of interpretations (Acc_Inte) and their original representations (Acc_Ori). Taking ViT-B and ViT-L for examples, representations with higher accuracy (87.99% *vs.* 80.58%) can derive interpretations with higher ARR (98.70% *vs.* 97.42%), thereby yielding interpretable predictions with significantly improved accuracy (86.85% *vs.* 78.50%). This observation is consistent with our conclusions in Section 3.3. By leveraging the enhanced classifiability of pre-trained representations, we can employ their interpretations for predictions with performance closer to the original representations.

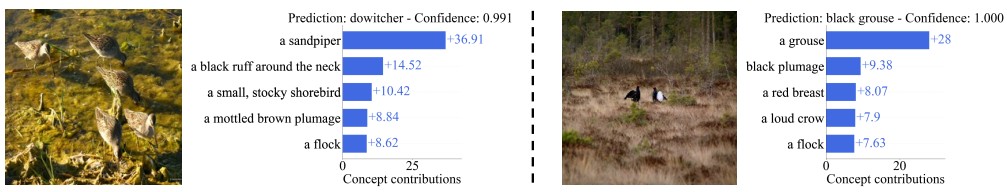

Figure 9: Visualizations of two correct samples of interpretable predictions

The visualization examples for interpretable predictions are shown in Figure 9, and more visualizations are available in Appendix A.3. These examples show the most important concepts with high absolute contribution value. For the image in Figure 9, recognizing this image as a "dowitcher" involves identifying "a sandpiper" as the most important concept.

Additionally, we show the impact of sparsity on the interpretations in Figure 10 by providing an overview of the significant concept-class relationships. Given low sparsity ratios, these relationships are low informative. Specifically, a single class is correlated to concepts with comparable contribution values, and similarly, a concept correlates to classes with comparable contributions. As a result, the concepts are indistinguishable to classes, and vice versa, failing to provide clear knowledge for humans about the crucial concept in predictions and classification rules learned by

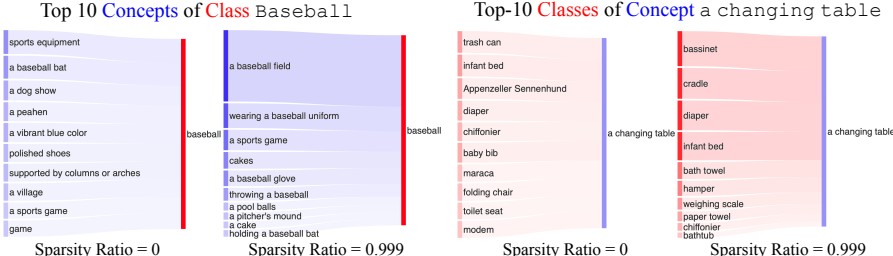

Figure 10: Visualization of the average contribution of concepts to classes, showcasing how the sparsity ratio affects the quality of interpretations through the entropy of concept contributions.

the models. By increasing the sparsity ratio, the model tends to correlate a class to a smaller number of key concepts and provide clearer insights into its decision-making process.

## 5 RELATED WORK

**Post-hoc Interpretation Methods.** Post-hoc methods have been widely studied for interpreting model predictions by mapping the model's representation to human-understandable domains (*e.g.*, image pixels, words*, etc.*). To interpret representations, previous work attempts to decouple representations into human-understandable elements such as concept vectors (Ghorbani et al., 2019; Oikarinen et al., 2023), explanation graph (Zhang et al., 2017; 2018) or decision tree (Zhang et al., 2019; Bai et al., 2019). Inspired by methods that correlate representations with concepts, we extract interpretable semantics embedded within representations for their interpretability measurement.

**Interpretability Quantification.** Interpretability describes the extent to which humans are capable of understanding (Longo et al., 2020). As a subjective metric, interpretability can be impacted by many factors, including causality (Holzinger et al., 2019), faithfulness (Khakzar et al., 2022), and sparsity (Oikarinen et al., 2023). Existing metrics are based on heuristic approaches that compare the relevance attributed to the different features with the expectation of an observer (Bau et al., 2017), a domain expert (Neves et al., 2021), or calculate the homogeneity score between interpretations and ground-truth labels (Zarlenga et al., 2022). Different from the above metrics designed for model interpretations, we focus on pre-trained representations and quantify their interpretability by their accuracy retention ability when predicting solely based on interpretable semantics.

**Interpretability-Classifiability Relation.** The conflicting relationship between interpretability and classifiability has been observed in interpretability-oriented methods and studied for years (Baryannis et al., 2019; Rudin et al., 2022). It says that models imposed with stronger interpretability constraints or inductive bias demonstrate weaker classification accuracy (Mori & Uchihira, 2019; Mahinpei et al., 2021; Zarlenga et al., 2022; Dombrowski et al., 2023). Existing work tries to overcome this issue by the design of interpretable architectures (Zarlenga et al., 2022) or feature engineering (Gosiewska et al., 2021) and make slow progress. In contrast to previous investigations regarding interpretability-oriented methods, we focus on classifiability-oriented representations and indicate that classifiability and interpretability are not conflicting but positively correlated.

## 6 CONCLUSION

We explored the possibility of achieving high interpretability and classifiability simultaneously by identifying interpretable semantics within pre-trained representations. First, we proposed the IIS to measure the representation interpretability by the maintenance of task-relevant semantics in interpretations. Then, we investigated the interpretability-classifiability relationship of pre-trained representations. Extensive experiments on multiple datasets indicate that classifiability and interpretability are positively correlated. By enhancing representation interpretability, we improve the classifiability of pre-trained representations. By improving representation classifiability, we offer interpretable predictions based on interpretations with more comparable accuracy to the original representations. For future work, we will provide a theoretical analysis of the interpretability-classifiability relationship and take more factors (*e.g.,* trustworthiness, structural entropy of representations) to refine IIS, and seek to develop an interpretable training paradigm for vision-language foundation models.

## REPRODUCIBILITY STATEMENT

To ensure the reproducibility of our work, we provide comprehensive details on datasets, pre-trained models, and concept library construction in Section 3.1. Details of training strategies and IIS computing can be found in Appendix A.1. Source code has been submitted as supplementary materials.

## ACKNOWLEDGEMENT

This work was supported in part by the National Key R&D Program of China under Grant 2023YFC2508704, in part by the National Natural Science Foundation of China: 62236008, 62022083, U21B2038, 62306092 and 62441232, and in part by the Fundamental Research Funds for the Central Universities. The authors would like to thank Zhengqi Pei, Yue Wu, and the anonymous reviewers for their constructive suggestions that improved this manuscript.

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

Table A6: Configurations of Selected Pre-trained Models.

| Model | Interpolation | Resize | Input Size | Pretrain |
|---|---|---|---|---|
| ResNet-18 | Bilinear | 256×256 | 224×224 | ImageNet1K_V1 |
| ResNet-34 | Bilinear | 256×256 | 224×224 | ImageNet1K_V1 |
| ResNet-50 | Bilinear | 256×256 | 224×224 | ImageNet1K_V1 |
| ResNet-101 | Bilinear | 256×256 | 224×224 | ImageNet1K_V1 |
| ResNet-152 | Bilinear | 256×256 | 224×224 | ImageNet1K_V1 |
| ViT-B-16 | Bilinear | 256×256 | 224×224 | ImageNet1K_V1 |
| ViT-L-16 | Bilinear | 512×512 | 512×512 | ImageNet1K_SWAG_E2E_V1 |
| Swin-T | Bicubic | 246×246 | 224×224 | ImageNet1K_V1 |
| Swin-S | Bicubic | 246×246 | 224×224 | ImageNet1K_V1 |
| Swin-B | Bicubic | 238×238 | 224×224 | ImageNet1K_V1 |
| ConvNeXt-T | Bilinear | 236×236 | 224×224 | ImageNet1K_V1 |
| ConvNeXt-S | Bilinear | 230×230 | 224×224 | ImageNet1K_V1 |
| ConvNeXt-B | Bilinear | 232×232 | 224×224 | ImageNet1K_V1 |
| ConvNeXt-L | Bilinear | 232×232 | 224×224 | ImageNet1K_V1 |

# A APPENDIX

## A.1 IMPLEMENTATION DETAILS

### A.1.1 CONCEPT LIBRARY CONSTRUCTION

**Visual Concepts.** Given an image classification dataset with $N$ classes, we randomly sample 10 images for each class. For these images, we extract segments or patches from them to build visual concepts. For patches, we resize each image to dimensions of $224 \times 224$. Then, we split each resized image into $56 \times 56$ patches. For segments, we apply the SLIC (Achanta et al., 2012) algorithm to extract segments unsupervisedly. We perform the segmentation process three times, resulting in 10, 20, and 30 segments respectively, to capture segments at different scales. To avoid redundancy, we eliminate segments with significant overlap at the same scale. During the experiment, we observed high redundancy among these segments/patches. Therefore, we randomly select $H = 20$ segments/patches for each class. These selected patches or segments are then aggregated to form $M$ visual concepts. Here we take the patch as an example for simplicity.

For *Prototype* concept library, we randomly select $\lceil \frac{M}{N} \rceil$ patches for each class.

For *Cluster* concept library, we aggregate all patches into $M$ visual concepts by clustering based on their latent vectors using the K-Means algorithm.

The *End2End* concept library is specific to a pre-trained model $f$. We design a learnable method to aggregate the patches into concepts based on their latent vectors generated by $f$. The learning objective is to obtain concepts that maximize the classification accuracy built upon $f$. Specifically, for the concatenation of latent vectors of all patches $\mathbf{C}^p \in \mathbb{R}^{D \times HN}$, we assign these patches to $M$ visual concepts by

$$\min_{\mathbf{Q}, g_{\text{cls}}} \mathbb{E}_{(x, \mathbf{y})}[\mathcal{L}(\mathbf{y}, g_{\text{cls}}(f(x)^\top \mathbf{C}_p \mathbf{Q}))], \tag{A9}$$

where $g_{\text{cls}} : \mathbb{R}^M \to \mathbb{R}^N$ depicts a linear classification layer, $\mathcal{L}$ means the cross-entropy loss function, $\mathbf{Q} = [\mathbf{q}_1, \mathbf{q}_2, ..., \mathbf{q}_N] \in \{0, 1\}^{HN \times M}$ is a learnable transportation matrix responsible for assigning patches to visual concepts. Each column $\mathbf{q}_n \in \{0, 1\}^M$ is a one-hot vector that assigns the $n$-th patch to one of the $M$ concepts. In practice, the discrete constraints of $Q$ are enforced through Gumbel-Softmax with Straight Estimator (Jang et al., 2017).

**Textual Concepts.** Following previous work (Oikarinen et al., 2023), the textual concepts are automatically extracted for each class. The prompts are as follows:

- List the most important features for recognizing something as a {class}:
- List the things most commonly seen around a {class}:
- Give superclasses for the word {class}:

Table A7: Sparsity Selection Strategies for different concept libraries.

| Dataset | Concept Library | Concept Num. | Selected Sparsity Ratios |
|---------|-----------------|--------------|--------------------------|
| ImageNet | Prototype | 200 | $\{0, 0.1, 0.3, 0.5, 0.7, 0.9, 0.95, 0.98\}$ |
| | Cluster | 200 | $\{0, 0.1, 0.3, 0.5, 0.7, 0.9, 0.95, 0.98\}$ |
| | End2End | 200 | $\{0, 0.1, 0.3, 0.5, 0.7, 0.9, 0.95, 0.98\}$ |
| | Text | 4751 | $\{0, 0.9, 0.99, 0.995, 0.997, 0.999\}$ |
| CUB-200 | Text | 370 | $\{0, 0.5, 0.7, 0.9, 0.99, 0.995\}$ |
| CIFAR-10 | Text | 143 | $\{0, 0.5, 0.7, 0.9, 0.95, 0.97\}$ |
| CIFAR-100 | Text | 892 | $\{0, 0.5, 0.7, 0.9, 0.95, 0.97\}$ |

Moreover, the extracted concepts are filtered. First, concepts with a lot of words are removed to make them easy to visualize and understand. Second, concepts with large similarities with target classes are removed to prevent trivial explanations. Third, the retention of duplicate concepts (cosine similarity greater than 0.9) is limited to only one instance. Finally, concepts not presented in the training images (cosine similarity less than 0.4) are removed.

The number of concepts for different datasets is as follows: 143 for CIFAR-10, 892 for CIFAR-100, 370 for CUB-200, and 4751 for ImageNet1K. We use CLIP (ViT-B-16) (Radford et al., 2021) to generate soft concept labels for all datasets.

Specifically, we use the same configurations as Torchvision for the selected pre-trained models presented in Table A6. All models are pre-trained on ImageNet, except for ViT-L, which is pre-trained using SWAG (Singh et al., 2022) and fine-tuned on ImageNet.

### A.1.2 SPARSITY RATIO SELECTION

Practically, we compute IIS by selecting a series of sparsity ratios $s$. The selection strategy of $s$ corresponds to the concept library. It should be noted that we avoid selecting an excessively high sparsity ratio resulting in extremely sparse interpretations that turn out to be confusing to humans (*e.g.*, a single concept). The specific selection strategies are shown in Table A7.

### A.1.3 TRAINING STRATEGIES

During the training process in Section 2.2, the pre-trained model $f$ is frozen. For Prototype and Cluster concept libraries, only the classifier $g_{\text{cls}}$ is trainable. For End2End concept library, the projection from representation space to concept space $g_{\mathcal{C}}$ and classifier $g_{\text{cls}}$ are trainable. For Text concept library, the training process is divided into two stages. In the first stage, the concept vectors **C** are obtained through Equation 2. In the second stage, only the classifier $g_{\text{cls}}$ is trainable.

We present the hyperparameters during training in Table A8. For each model, we train three times using multiple learning rates of $\{0.1, 0.01, 0.001\}$ and select the result with the highest accuracy, as different models may perform better with different learning rates. For visual concepts (Prototype, Cluster, and End2End), we utilize the Adam optimizer and the exponential scheduler with a decay rate of $0.99$. For textual concepts (Text), we utilize the SGD optimizer without schedulers.

### A.1.4 IIS MAXIMIZATION

For the fine-tuning process with IIS maximization as the objective, we utilize the sparsity ratio $s = 0.1$ and vector number $M = 200$ to compute the simplified IIS. The pre-trained models undergo fine-tuning for 200 epochs (with the first 20 epochs to warmup), utilizing a batch size of 128. The finetuning process incorporates the AdamW optimizer with betas set to $(0.9, 0.999)$, a momentum of 0.9, a cosine decay learning rate scheduler, an initial learning rate of $3e - 4$, and a weight decay of 0.3. Additional techniques such as label smoothing $(0.1)$ and cutmix $(0.2)$ are also employed. Beyond these, no further techniques are applied.

### A.2 ADDITIONAL EXPERIMENTAL RESULTS

Table A8: Hyperparameters for CBM Training

| Dataset | Concept Libraries | Learning Rate | Epoch | Optimizer | Scheduler | Batch Size |
|---|---|---|---|---|---|---|
| ImageNet | Prototype | $\{0.1, 0.01, 0.001\}$ | 30 | Adam | EXP | 1024 |
| | Cluster | $\{0.1, 0.01, 0.001\}$ | 30 | Adam | EXP | 1024 |
| | End2End | $\{0.1, 0.01, 0.001\}$ | 30 | Adam | EXP | 1024 |
| | Text | $\{0.1, 0.01, 0.001\}$ | 30 | SGD | NAN | 256 |
| CUB-200 | Text | $\{0.1, 0.01, 0.001\}$ | 100 | SGD | NAN | 256 |
| CIFAR-10 | Text | $\{0.1, 0.01, 0.001\}$ | 100 | SGD | NAN | 256 |
| CIFAR-100 | Text | $\{0.1, 0.01, 0.001\}$ | 100 | SGD | NAN | 256 |

Table A9: IIS-accuracy relationship on CUB-200 dataset with different sparsification mechanisms.

| Model | Acc. | Origin | Descending | Clustering | HardThres |
|---|---|---|---|---|---|
| ResNet-18 | 63.39 | 0.695 | 0.284 | 0.664 | 0.769 |
| RseNet-34 | 64.22 | 0.719 | 0.298 | 0.675 | 0.774 |
| RseNet-50 | 67.01 | 0.766 | 0.340 | 0.721 | 0.809 |
| ResNet-101 | 67.13 | 0.776 | 0.352 | 0.732 | 0.816 |
| RseNet-152 | 68.00 | 0.780 | 0.354 | 0.742 | 0.828 |

### A.2.1 INFLUENCE OF SPARSIFICATION MECHANISMS

In this section, we investigate the impact of different concept sparsification mechanisms on IIS. In addition to the strategy in Equation 4, we implemented three alternative approaches to investigate the relationship between interpretability and classifiability.

*(i) Descending.* We remove concepts in descending order based on their similarity to image representations. Specifically, given the similarity between image representations and concepts $\mathbf{x}^{\mathcal{C}} \in \mathbb{R}^M$ in Equation 3, we first invert it and then apply Equation 4 for sparsification. *(ii) Clustering.* We reduce the number of concepts by clustering rather than directly eliminating concepts. Specifically, given a sparsity ratio $s$, we cluster $M$ concepts to $\lceil (1 - s) \times M \rceil$ clusters with KMeans. Image representations are projected into these clusters for IIS computation. *(iii) HardThres.* We replace the soft threshold in Equation 4 with the hard threshold (*i.e.,* directly masking concepts with smaller value than the threshold).

As Table A9 shows, representations with higher classifiability still demonstrate higher IIS under different sparsification mechanisms. This phenomenon further demonstrates that the positive correlation between interpretability and classifiability is an inherent characteristic of pre-trained representations, rather than a consequence of specific sparsification methods.

### A.2.2 INFLUENCE OF TEXTUAL CONCEPT VECTOR ACQUISITION

Besides our methods to acquire textual concept vectors in Equation 2, LF-CBM (Oikarinen et al., 2023) has proposed the cos-cubed loss to learn concept vectors. For the similarities, the objective of both loss functions is to map a set of concept vectors $[\mathbf{c}_1, \mathbf{c}_2, ..., \mathbf{c}_M]$ in the representation space to textual concepts $[c_1, c_2, ..., c_M]$. For the dissimilarities, given the soft-label $\mathbf{y}_x = [y_x^{c_1}, y_x^{c_2}, ..., y_x^{c_M}]$ of image $x$ and the inner products between concept vectors and image representations $\mathbf{q}_x = [f(x)^\top \mathbf{c}_1, f(x)^\top \mathbf{c}_2, ..., f(x)^\top \mathbf{c}_M]$. The cos-cubed loss normalizes vectors $\mathbf{y}_x, \mathbf{q}_x$ and utilizes the inner product of their cubic powers as a similarity measure. In contrast, we directly employ the mean square error based on the normalized vectors.

Additionally, we conduct experiments with cos-cubed loss as Table A10 shown. Replacing Equation 2 with cos-cubed loss does not impact the positive correlation between IIS and classifiability, which verifies the generality of our conclusion.

### A.2.3 EXPERIMENTS ON ADDITIONAL DATASETS & MORE COMPARISONS

Table A10: IIS-accuracy relationship on CUB with the cos-cubed loss for concept vector acquisition.

| Metric | ResNet-18 | ResNet-34 | ResNet-50 | ResNet-101 | ResNet-152 |
|--------|-----------|-----------|-----------|------------|------------|
| IIS | 0.756 | 0.774 | 0.789 | 0.802 | 0.805 |
| Acc@1 | 63.39 | 64.22 | 67.01 | 67.13 | 68.00 |

Table A11: IIS of ResNet-18/50/152 on different datasets. Experiments are conducted with textual concept libraries from LaBo.

| Model | UCF101 | DTD | HAM |
|-------|--------|-----|-----|
| ResNet-18 | 0.949 | 0.934 | 0.912 |
| ResNet-50 | 0.967 | 0.940 | 0.914 |
| ResNet-152 | 0.976 | 0.957 | 0.915 |

Table A12: Performance comparisons on different datasets. Experiments are conducted with textual concept libraries from LaBo.

| Model | UCF101 | DTD | HAM |
|-------|--------|-----|-----|
| LaBo | 97.68 | 76.86 | 81.40 |
| Ours | **97.79** | **77.22** | **81.90** |

Table A13: Performance comparisons with LF-CBM on different representations.

| Model | ImageNet | CUB |
|-------|----------|-----|
| LF-CBM (ViT-B) | 77.94 | 68.97 |
| LF-CBM (ViT-L) | 85.52 | 82.83 |
| Ours (ViT-B) | 78.50 | 69.14 |
| Ours (ViT-L) | 86.85 | 83.51 |

**Interpretability-Classifiability Relationship on Additional Datasets.** To further illustrate the generality of our conclusions in Section 3.2, we conduct experimental results on addition datasets including UCF-101 (Soomro, 2012), DTD (Cimpoi et al., 2014) and HAM10000 (Tschandl et al., 2018). In Table A11, we further verify the positive relationship between interpretability and classifiability on three datasets, indicating the strong generality of our conclusions.

**More Performance Comparisons.** Performance comparisons with LaBo (Yang et al., 2023) on 3 additional datasets (Soomro, 2012; Cimpoi et al., 2014; Tschandl et al., 2018) are provided in Table A12. Our interpretable predictions demonstrate superior performance compared to LaBo. Furthermore, Table A13 presents the comparisons with LF-CBM (Oikarinen et al., 2023), a framework for providing interpretable predictions based on arbitrary vision representations. Our method outperforms LF-CBM using different pre-trained representations.

**Effects of Vision-Language Pre-training on Classifiability and Interpretability.** In Table A15, we compute the IIS of ViT-B and CLIP-ViT-B on the CUB-200 dataset. The experimental results remain consistent with our investigations in Section 3.2, i.e., representations with higher classifiability exhibit enhanced interpretability. Representations pre-trained with large-scale vision-language datasets have stronger generalization ability than representations of the standard ViT, thereby resulting in improved classifiability on this dataset and subsequently enhancing their interpretability.

### A.2.4 RELATIONSHIP BETWEEN IIS AND SEGMENTATION PERFORMANCE

The classification task serves as a fundamental component of vision pre-training. Enhanced classifiability in representations often leads to improved performance on downstream vision tasks (Liu et al., 2021). For example, therefore, we can derive a positive correlation between IIS and segmentation performance. Given the pre-trained representations of ResNet50, we perform two sets of comparative experiments. (i) We directly fine-tune the representations on segmentation tasks (Origin). (ii) We first adjust the representations through IIS maximization in Equation 8 and then fine-tune them on segmentation tasks (Ours). As Table A14 shows, compared to original representations, enhancing the IIS of representations leads to an improvement in their classifiability (Acc.). The improvement leads to an increase in the segmentation performance (mIoU).

### A.2.5 CONCEPT ACTIVATION VISUALIZATIONS

We visualize the activation locations of each concept through Grad-CAM and compare them with the input images. As Figure A11 shows, the activation regions of the concept are related to the concept

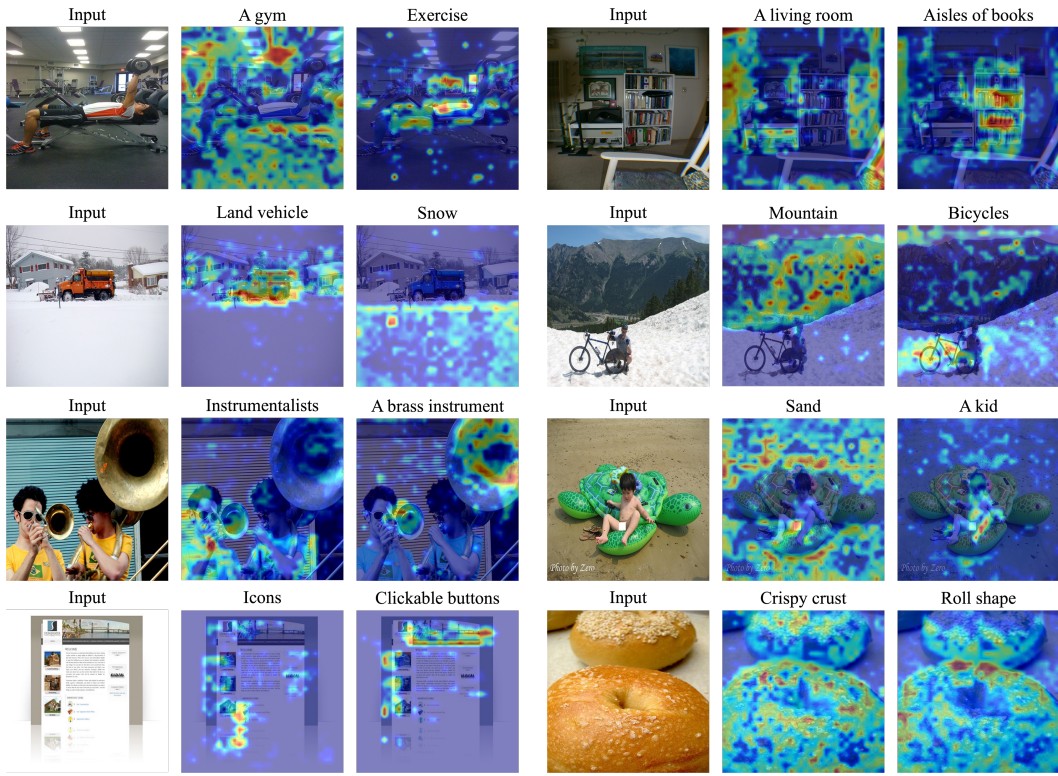

Figure A11: Concept activation visualizations with pre-trained representations of ViT-L.

Table A14: Relation between IIS, accuracy on ImageNet-1k and mIoU on ADE20k.

| Model | IIS | Acc. | mIoU |
|---|---|---|---|
| Origin | 0.953 | 75.66 | 40.60 |
| Ours | 0.958 | 77.36 | 42.35 |

Table A15: Comparisons between standard ViT and CLIP-ViT on CUB.

| Model | IIS | Acc@1 |
|---|---|---|
| ViT-B | 0.764 | 75.9 |
| CLIP-ViT-B | 0.808 | 80.6 |

Table A16: Performance comparisons with DEAL on ImageNet-1k.

| Model | CLIP-RN50 | CLIP-ViT-B/32 |
|---|---|---|
| DEAL | 70.0 | 70.8 |
| Ours | **70.8** | **73.0** |

semantic, thereby qualitatively supporting the trustworthiness of leveraging powerful pre-trained representations (*e.g.,* ViT-L) for interpretable predictions.

### A.2.6 SPARSITY-ARR CURVES

In this section, we show the sparsity-ARR curves of more pre-trained representations in Figure A13, to better support our conclusions in Section 3.3. We observe a similar phenomenon to Figure 6, demonstrating that for representations with fixed accuracy, the information loss increases when projecting pre-trained representations into more human-understandable patterns.

### A.2.7 ABLATION STUDIES OF VISUAL CONCEPTS

We conducted verification experiments to confirm the consistency of our findings from Figure 4 in the main text. These experiments encompassed different types of visual elements, including patches and segments. In Table A17, we present the relationship between representation accuracy and IIS computed using the End2End concept library aggregated from patches and segments respectively. The experimental results show the same phenomenon as Figure 4 in the main text. Moreover, we observe that segments achieve a higher IIS than patches.

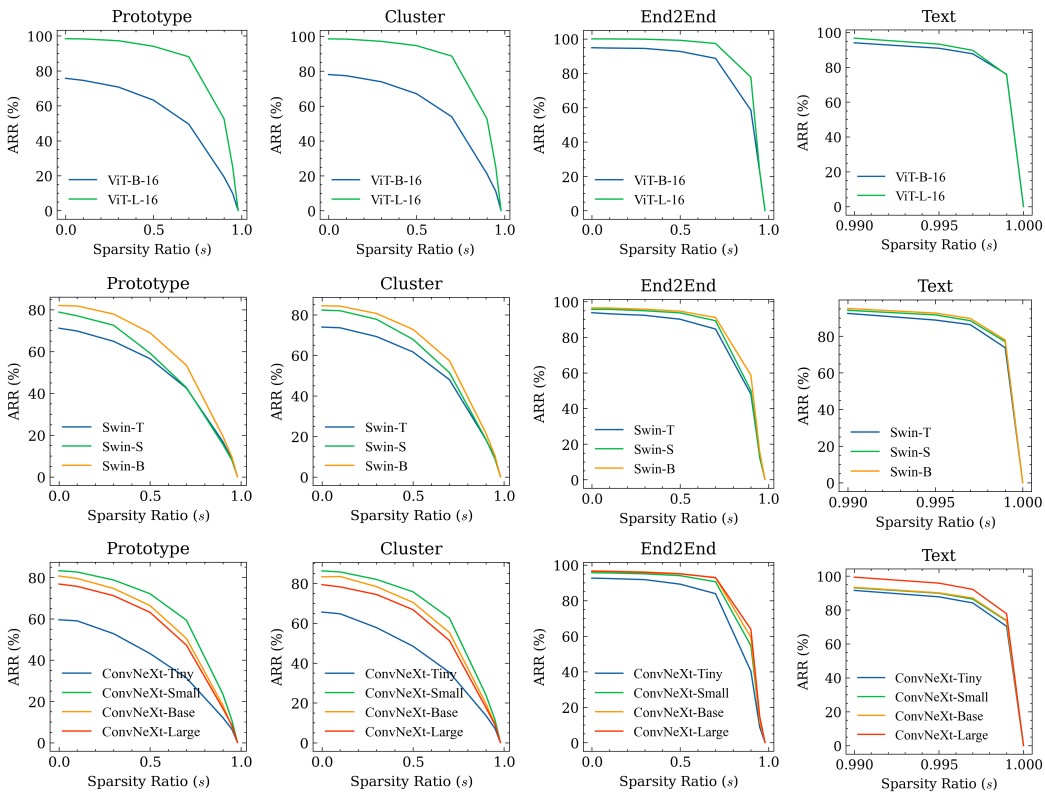

Figure A12: More experiments about the sparsity-ARR curve of different pre-trained representations on ImageNet with four types of concept libraries.

Table A17: Ablation studies of different visual elements (Patches and Segments)

| Visual Elements | ResNet-18 | ResNet-34 | ResNet-50 | ResNet-101 | ResNet-152 |
|---|---|---|---|---|---|
| Patches | 0.6833 | 0.7021 | 0.7210 | 0.7231 | 0.7654 |
| Segments | 0.6879 | 0.7043 | 0.7311 | 0.7386 | 0.7694 |

### A.2.8 RELATIONSHIP BETWEEN SIMPLIFIED IIS AND ORIGINAL IIS

In Equation 8, we maximize a simplified version of IIS to improve pre-trained representations. Experimental results in Table A18 show that during the IIS maximization process, the representation accuracy and original IIS increase with the simplified version of IIS. This indicates that the maximization of the simplified IIS can improve the original IIS.

Table A18: The accuracy (Acc.), simplified IIS, and original IIS of ResNet-50 finetuned on ImageNet with the IIS (Simplified) maximization as training objective.

| Epoch | 0 | 40 | 200 |
|---|---|---|---|
| Acc. | 0.7566 | 0.7644 | 0.7736 |
| IIS (Simplified) | 0.9654 | 0.9776 | 0.9805 |
| IIS (Original) | 0.9533 | 0.9573 | 0.9583 |

## A.3 ADDITIONAL VISUALIZATION OF INTERPRETABLE PREDICTIONS

We provide more visualizations of interpretable predictions based on textual (Figure A14) and visual (Figure A15) concepts on ImageNet. These were created using the same procedure as Figure 9. The concepts that have large contributions are relevant to the class. Additionally, Figure A16 showcases the model editing with human guidance by zeroing out the contribution of incorrect concepts such as concept bottleneck models (Zarlenga et al., 2022; Oikarinen et al., 2023).

## A.4 ADDITIONAL RESULTS ON VIDEO PRE-TRAINED MODELS

Building upon the observed positive correlation between interpretability and classifiability of image representations, we investigate whether this phenomenon extends to video representations on the Kinetics-400 dataset (Carreira & Zisserman, 2017). For video concept, we utilize the YOLOv5 (Jocher, 2020) pre-trained on COCO (Lin et al., 2014) to extract visual-grounding objects. These objects are aggregated as concepts with the strategy of the End2End concept library. By computing the IIS and prediction accuracy of representations from video pre-trained models (Carreira & Zisserman, 2017; Feichtenhofer et al., 2019; Bertasius et al., 2021; Fan et al., 2021; Tong et al., 2022; Wang et al., 2023; Li et al., 2023b; 2022b), we observe a similar phenomenon to that observed in image representations (*i.e.*, the positive correlation between IIS and accuracy).

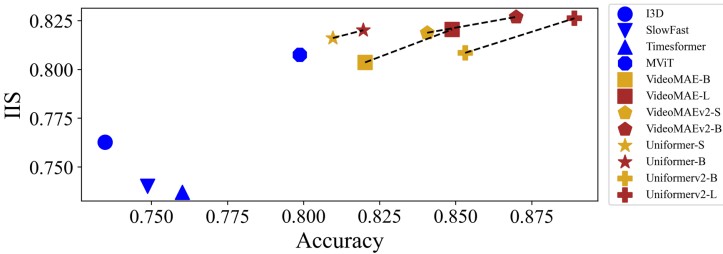

Figure A13: The relationship between accuracy and IIS for video pre-trained representations on the Kinetics-400 dataset. The IIS is computed based on the End2End concept library.

## A.5 DISCUSSION

### A.5.1 RELATIONSHIP BETWEEN IIS AND INTERPRETATION QUALITY METRICS.

IIS measures the representation interpretability through the accuracy comparison between representations and their corresponding interpretations. The formulation of IIS is similar to the faithfulness

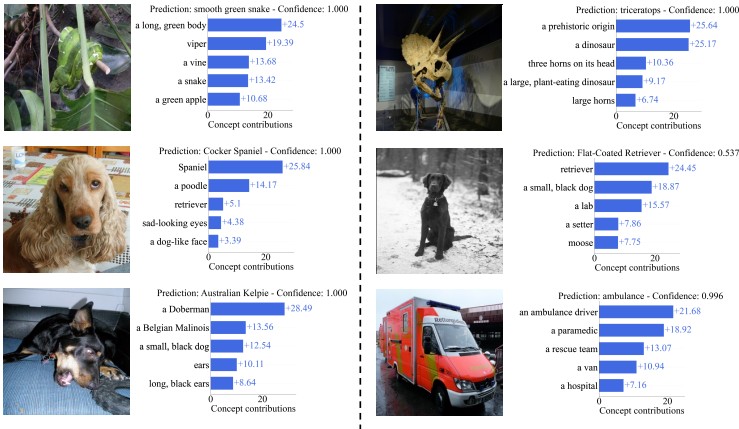

Figure A14: More visualizations of our interpretable predictions based on textual concepts.

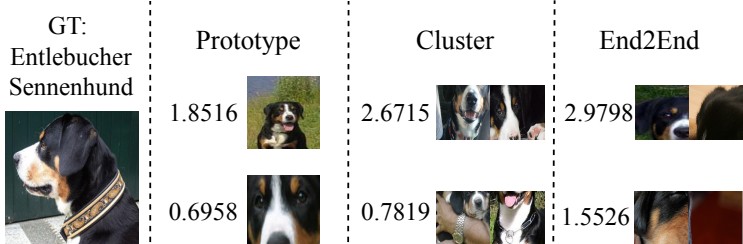

Figure A15: Visualizations of our interpretable predictions based on visual concepts.

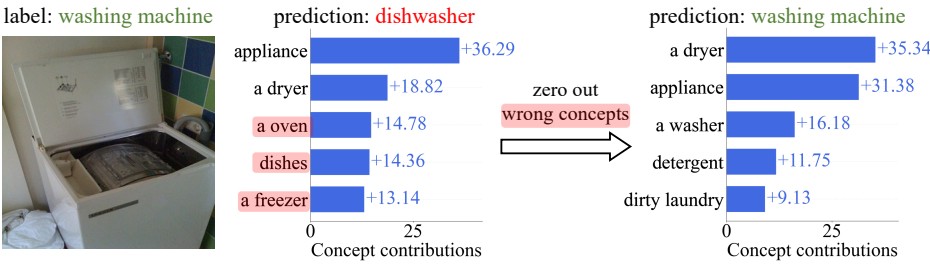

Figure A16: An example for model editing with human guidance. By zeroing out the contribution of incorrect concepts (*e.g.*, "a oven"), the prediction changes from "dishwasher" to "washing machine".

metric. This metric measures the predictive capacity of interpretations to evaluate how interpretations can meaningful and faithfully explain the model's predictions (Sarkar et al., 2022). However, the IIS and faithfulness metric are essentially different as follows:

- Different purposes: IIS is proposed to quantify the interpretability of pre-trained representations (comparing different representations), while the faithfulness metric is designed to measure the quality of interpretations (comparing different interpretation methods).

- Different definitions: the definition of IIS encompasses interpretation accuracy, representation accuracy, and interpretation sparsity, whereas the faithfulness metric solely focuses on interpretation accuracy.

### A.5.2 RELATIONSHIP BETWEEN IIS AND CONCEPT BOTTLENECK MODELS.

The prediction based on interpretations of IIS shares the same formulation as concept bottleneck models (CBMs) (Koh et al., 2020; Zarlenga et al., 2022; Oikarinen et al., 2023; Yan et al., 2023), which first identify fine-grained concepts for an input image and then make predictions based on identified concepts. We draw inspiration from concept bottleneck models in constructing concept libraries and the emphasis on interpretation sparsity. However, IIS primarily focuses on the interpretability measurement of pre-trained representations, while CBMs concentrate on improving the performance of concept-based predictions by designing new architecture (Zarlenga et al., 2022) and loss functions (Xu et al., 2023). Despite different focuses, our investigation contributes to the CBM community by revealing their strong compatibility with the development of pre-trained representations. This is because higher accuracy representations possess more interpretable semantics, which can be effectively captured by CBMs built upon them.

### A.5.3 COMPARISONS BETWEEN IIS AND VLG-CBM

VLG-CBM (Srivastava et al., 2024) also obtains accuracies of concept-based predictions with different sparsity levels. These accuracies are averaged to evaluate the ability of CBMs to provide accurate predictions with different numbers of concepts.

The sole similarity between IIS and this metric is assessing the accuracies of varying sparsity levels. The dissimilarities include three aspects. *(i)* Objective. The objective of IIS is to quantify the interpretability of arbitrary vision pre-trained representations. The average accuracy in VLG-CBM serves as a metric for evaluating the performance of CBMs, a specific type of interpretable model. *(ii)* Sparsification. Our sparsification mechanism can accurately control the number of concepts for prediction during forward propagation. VLG-CBM adjusts the number of concepts by manipulating the sparsity regularization strengths in the loss function, which requires complex strategies including gradually reducing the regularization strengths during training and post-training weight pruning. *(iii)* Metric Definition. IIS is defined as the area under the curve formed by the sparsity and accuracy. The metric in VLG-CBM is solely defined based on the average accuracy at different sparsity levels.

### A.5.4   Discussions with DEAL and Other Similar Methods

Recently, DEAL (Li et al., 2025) has observed that improving interpretability can have a positive impact on classification performance. For the performance, as Table A16 shows, our method outperforms DEAL on ImageNet with the same pre-trained representations. Moreover, our methods are different from DEAL in three aspects.

*(i)* Training strategy. DEAL is specifically designed for the CLIP architecture and employs contrastive learning methods to fine-tune the entire model. Our method can be applied to arbitrary architectures and only fine-tune the visual encoder. *(ii)* Prediction mechanism. DEAL predicts based on the average embedding similarity between the image and concepts within the category. Our method inputs the image representations to a prediction head without the text encoder. *(iii)* Interpretability-classifiability relationship. DEAL solely observed a qualitative impact of interpretability on classifiability in CLIP. In comparison, we find the mutually promoting relationship between both factors on a wide range of models in a quantitative manner.

Additionally, apart from DEAL, other approaches (Zhao et al., 2024; Gandelsman et al., 2023) have also demonstrated an enhancement in classifiability through the improvement of interpretability. Compared with these methods, our method may have similar motivations in exploring the relationship between interpretability and classifiability.

However, our contributions are different from these methods. *(i) Methods*. In contrast to these papers that focus on designing interpretation methods, we propose "a general metric" of representation interpretability to "quantitatively investigate" the interpretability-classifiability relationship. *(i) Investigation results*. While previous approaches have observed that enhancing interpretability improves classifiability (interpretability→classifiability), we have discovered "a mutually promoting relationship" (interpretability↔classifiability). Nevertheless, incorporating interpretability enhancements from these methods into IIS raises an interesting direction for further explorations.

### A.5.5   Limitation

Despite revealing the mutual-promoting relationship between representation interpretability and accuracy, investigations of the underlying mechanism leading to this relationship have not been conducted. In addition, we fine-tune pre-trained representations by maximizing just a simplified version of IIS to improve their accuracy. For a deeper understanding of the consistency between interpretability and accuracy, we anticipate future work to investigate the underlying mechanisms of our observations, and better integrate IIS into the training process of pre-trained models.

