# OpenReview forum: "Enhancing Pre-trained Representation Classifiability can Boost its Interpretability"
_ICLR.cc/2025/Conference — ICLR 2025 Spotlight_

### Official Review · Reviewer_wUp3 · 2024-10-31

**Soundness:** 2
**Presentation:** 3
**Contribution:** 3
**Rating:** 6
**Confidence:** 4

**Summary:**

The paper proposed a novel representation interpretability metric called IIS. A series of experiments are conducted which shows positive correlation between IIS and classifierbility of the representation. This observation is applied in improving classifierbility of representation and reduce accuracy loss for interpretation.

**Strengths:**

1. The proposed IIS metric provides an interesting way to measure representation interpretability.
2. The observation on relation between interpretability and classifiability is novel and backed by extensive experiments.

**Weaknesses:**

1. Though the idea of IIS intuitively makes sense, more theoretical discussion on this metric is needed. For example, suppose there is a random representation where the prediction accuracy no better than random guess. In this case, the representation will have low classifiability while high IIS (ARR = 1 as the accuracy will not further decrease after sparsification).

**Questions:**

1. How is Eq. (2) derived? It looks like the cos-cubed loss in [1]. More discussion on the details is needed.
2. In sparsification in Eq.(4), besides zeroing out elements, the remaining elements are also adjusted by their distance to threshold. Is there any justification to this sparsification? How will the IIS change for different sparsification method (e.g. only masking out small absolute value elements without changing others).
3. In Table 4, how is the accuracy for your method calculated? What is the sparity ratio $s$ used here?

[1] Oikarinen, Tuomas, et al. "Label-free concept bottleneck models." arXiv preprint arXiv:2304.06129 (2023).

---

> ### Author Response · Authors · 2024-11-20
> **Official Comment by Authors (1/2)**
>
> Thank you very much for the valuable and insightful feedback, which has greatly contributed to refining our manuscripts. We hope the following responses can address your concerns.
>
> ## W1: IIS for representations with extremely low classifiability
>
> Your comments are valuable for us to further improve the theoretical properties of IIS.
>
> **IIS is designed for pre-trained representations**. We would like to clarify that the purpose of IIS is to investigate the interpretability-classifiability relationship of classifiability-oriented ``pre-trained`` representations. Therefore, we did not consider representations with extremely low classifiability in the design of IIS.
>
> **Taking random representations into consideration**. We recognize and appreciate your suggestions to complete the IIS definition and will make further improvements in future work. For example, we can incorporate structural entropy [1] into IIS and assign low IIS values for high entropy representations (such as random representations).
>
> **Revision**. We have mentioned this phenomenon in L312-314 and added the target of refining IIS by considering the structural entropy in Section 6 of the revision.
>
> ## Q1: Derivation of Equation 2
>
> Thank you very much for your careful feedback, which helped us identify and correct the typo in Equation 2.
>
> **Fixing the typo**. We inadvertently wrote the quadratic term as a cubic term in the original manuscript, and this has been addressed in the revised version.
>
> **Discussion with cos-cubed loss**. The relationship between Equation 2 and the cos-cubed loss in [2] can be summarized as:
> - **Similarities**. The objective of both loss functions is to map a set of concept vectors $[\mathbf{c}_1, \mathbf{c}_2,...,\mathbf{c}_M]$ in the representation space to textual concepts $[c_1, c_2,...,c_M]$.
>
> - **Dissimilarities**. Given the soft-label $\mathbf{y}_x=[y_x^{c_1},y_x^{c_2},...,y_x^{c_M}]$ of image $x$ and the inner products between concept vectors and image representations $\mathbf{q}_x=[f(x)^\top \mathbf{c}_1, f(x)^\top \mathbf{c}_2,...,f(x)^\top \mathbf{c}_M]$. The cos-cubed loss normalizes vectors $\mathbf{y}_x$ and $\mathbf{q}_x$, and utilizes the inner product of their cubic powers as a similarity measure. In contrast, we directly employ the mean square error based on the normalized vectors.
>
> - **Experiments with cos-cubed loss**. We conduct additional experiments with cos-cubed loss. As the following table shows, ``replacing Equation 2 with cos-cubed loss does not impact the positive correlation between IIS and classifiability``, which verifies the generality of our conclusion.
>
>     | Model | ResNet-18 | ResNet-34 | ResNet-50 | ResNet-101 | ResNet-152 |
>     | ----- | --------- | --------- | --------- | ---------- | ---------- |
>     | IIS   | 0.756     | 0.774     | 0.789     | 0.802      | 0.805      |
>     | Acc@1 | 63.39     | 64.22     | 67.01     | 67.13      | 68.00      |
>
> **Revision**: We have fixed the typo and added the discussion of the relationship between Equation 2 and cos-cubed loss in Section A.2.2.

---

> ### Author Response · Authors · 2024-11-20
> **Official Comment by Authors (2/2)**
>
> ## Q2: Sparsify concepts by only masking the small absolute values
>
> Thank you for your insightful suggestions for improving the sparsification process.
>
> **Justification**: The adjustment of the remaining elements by their distance to the threshold follows the soft-threshold function, which is widely employed for sparsification [3].
>
> **Experiments**: Following your suggestions, we explore the influence of ``only masking out small absolute value elements without changing others" on IIS (masking-only). Experiments are conducted with the ResNet series on CUB-200.
>
> | Model        | ResNet-18 | ResNet-34 | ResNet-50 | ResNet-101 | ResNet-152 |
> | ------------ | --------- | --------- | --------- | ---------- | ---------- |
> | IIS (masking-only) | 0.769     | 0.774     | 0.809     | 0.816      | 0.828      |
> | IIS (w/ adjustment) | 0.695     | 0.719     | 0.766     | 0.776      | 0.780      |
> | Acc@1        | 63.39     | 64.22     | 67.01     | 67.13      | 68.00      |
>
> **Analysis**. First, we surprisingly observe an increase in IIS with only the masking. This implies that using hard thresholds can improve the accuracy of interpretable predictions in comparison to the commonly employed soft thresholds for sparsification. Second, it is noteworthy that ``interpretability remains positively correlated with classifiability within the same sparsification method``. Our conclusions are not constrained to certain sparsification methods.
>
> **Revision**: Based on the aforementioned results, we have provided a detailed discussion in Section A.2.1 regarding the impact of different sparsification methods on IIS values.
>
> ## Q3: Details of Table 4
>
> **Accuracy calculation**. The accuracy in Table 4 is calculated based on the predictions of linear classifier $g_{\text {cls}}$ in L206-210 of the revision.
>
> **Sparsity ratios used in experiments**. We leverage the sparsification mechanism in Equation 4 to control the number of concepts available for prediction. The sparsity ratio $s$ is set as 0.5 and 0 for models with 123 concepts and 247 concepts, respectively.
>
> **Revision**. We have mentioned the usage of classifier $g_{\text {cls}}$ in Section 4.2 of the revision.
>
> [1] Wu, J., et al. (2022). Structural entropy guided graph hierarchical pooling. ICML.
>
> [2] Oikarinen, T., et al. (2023). Label-free concept bottleneck models. ICLR.
>
> [3] Chen, Y., et al. (2023). A unified framework for soft threshold pruning. arXiv.

---

> > ### Comment · Reviewer_wUp3 · 2024-11-22
> >
> > Thank you for the detailed response! This addressed my concerns. I will keep my score.

---

> > > ### Author Response · Authors · 2024-11-23
> > >
> > > Dear reviewer, thank you very much for your positive feedback and support. Your suggestions have helped us improve our work. We are glad that our responses addressed your concerns.

---

### Official Review · Reviewer_511b · 2024-11-01

**Soundness:** 3
**Presentation:** 3
**Contribution:** 3
**Rating:** 8
**Confidence:** 4

**Summary:**

The authors propose to quantify the interpretability of representations using the correlation with the ration of interpretable semantics. They propose the Inherent Interpretability Score to measure the interpretability and information loss of representations by checking how much a representation can retain the original accuracy after a projection into a sparse interpretable space and a sparsification process. Through this evaluation, they also analyze the correlation between classifiability and interpretability.

**Strengths:**

→ Proposition of an interpretability metric: I believe this is an important contribution to evaluate the interpretability of a model's representation. We know that given the subjectivity of interpretability, it is difficult to quantify this property. The authors present a consistent way to measure it using libraries of concepts and the projection of representations into concept space.

→ Testing of different concepts and validation through a variety of experiments: I appreciate the number of validation experiments, including 4 different concept libraries (visual and textual) and experiments to evaluate: accuracy vs. interpretability for different architectures, evolution through training epochs, sparsity ratio and representation dimension.

→ Two applications: The authors also provide two cases where their study can be applied, improving classifiability by maximizing interpretability and providing interpretable predictions with high accuracy.

**Weaknesses:**

→ I understand the positive correlation between interpretability and classifiability, but is it not too strong to assume that higher classifiability implies more interpretability in this part?:  “We discover the mutual promoting relation between the interpretability and classifiability
of the classifiability-oriented representations, indicating that improving the interpretability can further improve the classifiability, and representations with higher classifiability are more interpretable to preserve more task-relevant semantics in interpretations.”

As I understand it: There is a loss of information caused by the sparcity, which reduces the original accuracy.

→  I am not sure about the process of increasing sparsity. Could some of the zeroed concepts be some kind of sub-concept that gets diluted in the high dimensionality of the concept library? Or even the errors from obtaining the concept library? It would be interesting to see other ways to reduce the sparsity of these concepts.

→  In Figure 5, you can see that the model is interpretable at the beginning of training and then it loses interpretability. Why is the model interpretable when it knows nothing? The network was initialized with some pre-trained weights? Because if the weights are random, it makes no sense to be interpretable at the beginning. Do you know why this happens?

→  I think it would be important to include the standard deviation in Tables 1— 5.

→  I have not seen much difference in increasing sparsity to justify this sentence: “This indicates that the model is highly sensitive to the representation sparsity and should not employ excessively large sparsity constraints during fine-tuning.”

And another question about sparsity: if you are not able to provide a sparser representation, you are limiting the interpretability, right? Because 200 concepts with 0.1 sparsity is a lot to interpret I imagine.

→  I think Figure 1 would be more interesting if it were made with real data and real experiments (even with simplified results).

**Questions:**

I included some questions in the weaknesses section.

**Details Of Ethics Concerns:**

No concerns.

---

> ### Author Response · Authors · 2024-11-20
> **Official Comment by Authors (1/2)**
>
> Thank you very much for the detailed comments and suggestions, which have greatly benefited our research. We hope the following responses can address your concerns.
>
> ## W1: Clarification of L96-L99
>
> Thank you for your careful review. However, we may not fully understand your concern but we will do our best to address it. The clarification of this part is provided as follows.
>
> **Clarification**. The "higher classifiability implies more interpretability" is not an assumption but a `conclusion` of our experimental results in Figures 3, 4, and A12. We observe that the improvement in classifiability is accompanied by enhanced IIS.
> Enhanced IIS denotes higher interpretability, where representations can preserve more task-relevant semantics for interpretations.
>
>
> Please let us know if your concerns are related to the other points. We are standing by to address them.
>
> ## W2: Different methods to increase the sparsity of interpretations.
> We appreciate your valuable suggestions. We will clarify the order of concept removal and explore alternatives to reduce the number of concepts.
>
> **Concept removal orders**. For increasing sparsity, the concepts are removed in ascending order based on their correlation to the image representations in Equation 3. Concepts with limited correlation to image semantics are removed first.
>
> Regarding your assumption of removing similar sub-concepts, firstly, we have conducted deduplication when building concept libraries to avoid the existence of excessively similar sub-concepts. Secondly, the removal is performed based on the similarity between concepts and representations. Therefore, even if there are multiple similar concepts, they may be eliminated simultaneously due to their lack of relevance to image semantics.
>
> For your assumption of removing erroneous concepts, although we cannot guarantee the elimination of all errors, our sparsification strategy helps remove concepts that are absent in images through semantic similarities between concepts and images.
>
> **Alternatives for removing concepts**. Inspired by your suggestions, we design another way to reduce the number of concepts by aggregating them into a limited number of clusters. For IIS computation, the representations are projected into these cluster centers rather than sparsified concepts.
>
> - **Experiments**. The comparison results between our sparsification strategy and the cluster-based strategy are provided as follows.
>
>     | Model       | ResNet-18 | ResNet-34 | ResNet-50 | ResNet-101 | ResNet-152 |
>     | ----------- | --------- | --------- | --------- | ---------- | ---------- |
>     | IIS-Cluster | 0.664     | 0.675     | 0.721     | 0.732      | 0.742      |
>     | IIS         | 0.695     | 0.719     | 0.766     | 0.776      | 0.780      |
>     | Acc@1       | 63.39     | 64.22     | 67.01     | 67.13      | 68.00      |
>
> - **Analysis**. First, we still observe the positive correlation between classifiability and interpretability with the clustering strategy. Second, replacing our sparsification mechanism with clustering leads to the degradation in IIS. This is because our sparsification mechanism selects available concepts according to the input image while clustering first reduces the number of concepts and employs all the remaining concepts for images.
>
> **Revision**: Inspired by your question, we have provided a detailed discussion in Section A.2.1 regarding the impact of different sparsification methods on IIS values.

---

> ### Author Response · Authors · 2024-11-20
> **Official Comment by Authors (2/2)**
>
> ## W3: Discussion of IIS on models with random weights.
>
> Your comments are valuable for us to further improve the completeness of IIS.
>
> **Reasons for high IIS of random representations**. In Figure 5, the models are not initialized with pre-trained weights. The high IIS during the initial phase of pre-training is because the IIS is designed with the Accuracy Retention Ratio (ARR), which is the ratio of prediction accuracy
> based on interpretations to that of the original representations. Therefore, for a model with random weights, the IIS may be high because both interpretations and representations have similarly low accuracy.
>
> **Influences**. The primary objective of this paper is to investigate the interpretability-classifiability relationship of `pre-trained` representations, which consistently exhibit high accuracy. Therefore, this phenomenon does not affect the application of IIS in measuring the interpretability of pre-trained representations.
>
> **Inspirations**. However, we appreciate your findings of this phenomenon. Incorporating random representations can further complete the IIS definition. In future work, we plan to refine the definition of IIS by considering factors such as the structural entropy [1] of representations. For example, dividing IIS by entropy values can assign a lower IIS to random representations that have high entropy.
>
> **Revision**. We have incorporated reasons for the high IIS of random representations in L312-314 and added the target of refining IIS by considering the structural entropy in Section 6 of the revision.
>
> ## W4: Addition of the standard deviation
>
> We agree with your opinion that the standard deviation is important for validating the method's effectiveness.
>
> **Difficulty of providing standard deviation in large-scale datasets**. However, it is a prevailing issue in the field of computer vision that the standard deviation is often omitted due to the large scale of datasets and models [2,3]. Given the substantial size of ImageNet data, conducting repeated experiments on it demands excessive resources.
>
> **Adding Standard deviations of Table 4 in the revision**. Nevertheless, we make efforts to provide the standard deviation for Table 4 experiments with 5 different random seeds in the revision.
>
> | Method | #Concept | Acc@1 |
> | --- | --- | --- |
> | Ours (ViT-L) | 123 | 81.30$\pm$0.08 |
> | Ours (ViT-L) | 247 | 83.50$\pm$0.15 |
>
> ## W5: Insights of Table 3
>
> Thank you for your valuable suggestions that help improve our descriptions of experimental results.
>
> **Explanations of conclusions in Table 3**. The sparsity ratio $s$ plays a crucial role in balancing the loss of sparsity and classifiability during the fine-tuning process, i.e., larger $s$ means stronger sparsity constraints. As the experimental results show, the optimal results occur at $s=0.1$, a slight constraint strength. Any larger value of $s$ would result in performance degradation.
>
>
> **Your question about sparsity**. This is exactly what our findings can address. While sparsifying the less interpretable pre-trained representations with a large sparsity ratio leads to significant information loss, our investigations in Figure 7 demonstrate that as the classifiability of representations increases, they can be converted to sparser concepts with reduced information loss.
>
>
> **Revision**. For a better analysis of the experimental results, we have rewritten this sentence in L430-431 of the revision.
>
> ## W6: Re-drawing the Figure 1
>
> Thank you for your suggestion! We have re-drawed Figure 1 with real data and experiments in the revision.
>
> [1] Wu, J., et al. (2022). Structural entropy guided graph hierarchical pooling. ICML.
>
> [2] Yang, Y., et al. (2023). Language in a bottle: Language model guided concept bottlenecks for interpretable image classification. CVPR.
>
> [3] Oikarinen, T., et al. (2023). Label-free concept bottleneck models. ICLR.

---

> > ### Comment · Reviewer_511b · 2024-11-22
> >
> > Thank you for addressing my concerns and incorporating some suggestions. I will keep my score 8: accept, good paper.

---

> ### Author Response · Authors · 2024-11-22
>
> Dear Reviewer, we sincerely appreciate your support and recognition. We will follow your suggestions to implement our future work!

---

### Official Review · Reviewer_d5fZ · 2024-11-01

**Soundness:** 2
**Presentation:** 3
**Contribution:** 3
**Rating:** 6
**Confidence:** 4

**Summary:**

This paper introduces the Inherent Interpretability Score (IIS) to quantify interpretability within pre-trained visual models, establishing a positive correlation between classifiability and interpretability. The study shows that interpretability-oriented fine-tuning can further improve both interpretability and classifiability in vision models. Extensive evaluations across various models and datasets support these claims, positioning IIS as a metric for balancing classifiability and interpretability without sacrificing accuracy.

**Strengths:**

1. This paper proposes an innovative metric IIS, which provides a practical way to assess interpretability through classifiability and shows that pre-trained representations can achieve high interpretability and classifiability simultaneously.

2. Comprehensive experiments conducted have confirmed the positive correlations between interpretability and classifiability. In the meantime, authors have done thoroughly comparison experiments, showing the broad application direction in classification task.

**Weaknesses:**

1. The scope of this paper is limited to classifiability-oriented representations, where the fact that the representation can be applied to segmentation task is ignored.

2. There is limited discussion about the selection of textual concepts. The prompts used to generate concepts for each class appear to be basic, without addressing whether overlapping concepts might influence the results.

3. Compared to ECBM, proposed method achieves minimal improvement from 81.20% to 81.29%. Further comments will be listed below in Question section.

**Questions:**

1. While the paper uses segmentation as visual concepts, could the authors provide insights on whether IIS correlates with improvements in segmentation tasks?
2. In Table 4, compared with ECBM, your method improves the accuracy from 81.20% to 81.29% appears minimal. Could the authors provide a comparison between ECBM and your method when both use approximately 250 concepts?
3. Instead of concatenating both concepts together, could authors provide a comparison between using textual concepts only and visual concepts only?
4. While DEAL [1] also claimed that improving interpretability can have positive impact on classification performance too, could author provide a comparison results between DEAL and your method with regard to the classification accuracy improvement.

[1] DEAL: Disentangle and Localize Concept-level Explanations for VLMs http://arxiv.org/abs/2407.14412

---

> ### Author Response · Authors · 2024-11-20
> **Official Comment by Authors (1/2)**
>
> Thank you very much for your feedback, which has been of great help to our work. We hope the following responses can clarify the missing points and address your concerns.
>
> ## W1&Q1: Correlations between IIS and segmentation performance
> Thank you for your valuable comments.
> Based on the (i) positive correlation between segmentation performance and representation classifiability [1,2], (ii) positive correlation between representation classifiability and IIS (our manuscript), we can derive the positive correlation between IIS and segmentation performance.
>
> **Experiments**. Given the pre-trained representations of ResNet50, we perform two sets of comparative experiments. (i) We directly fine-tune the representations on segmentation tasks (Origin). (ii) We first adjust the representations through IIS maximization in Equation 8 and then fine-tune them on segmentation tasks (Ours).
>
> | Model      | IIS   | Acc.  | mIoU  |
> | ---------- | ----- | ----- | ----- |
> | Origin  | 0.953 | 75.66 | 40.60 |
> | Ours | 0.958 | 77.36 | 42.35 |
>
> **Analysis**. Compared to original representations, enhancing the IIS of representations leads to an improvement in their classifiability (Acc.). The improvement leads to an increase in the segmentation performance (mIoU).
>
> **Revision**. We have incorporated the experiments in Section A.2.4 of the revision.
>
> ## W2: Details of the textual concept extraction.
>
> Thank you for your careful review that helps us refine the illustration of implementation details.
>
> **Overlapping concepts have been removed**. Following previous work [3], given a concept, if there is another concept already in the concept set with a cosine similarity larger than 0.9, the concept will be removed. The similarity between concepts is measured in the embedding space of CLIP.
>
> **Revision**. We have added detailed descriptions of the textual concept extraction in Section A.1.1 of the revision.
>
>
> ## W3&Q2: More comparisons with ECBM.
>
> **Improvement compared to ECBM**. Our methods demonstrate superior performance by solely training the classification head with automatically generated concept annotations. In contrast, ECBM employs manual concept annotations to train the entire model.
>
> **Comparing to ECBM with the same concept library**. Following your suggestion, we re-implement the ECBM on our textual concept library.
>
> - **Experiments**. We provide the performance comparisons with ECBM on the CUB-200 dataset.
>
>     | Model | #Concepts | Acc@1 |
>     | ----- | --------- | ----- |
>     | ECBM  | 247       | 83.21 |
>     | Ours  | 247       | **83.50** |
>
> - **Analysis**. Within the same concept library, our method achieves better performance than ECBM by 0.29\%.
>
> **Revision**. We have incorporated comparisons with ECBM using the same concept library in Table 4 of the revision.
>
> ## Q3: The utilization of different concept types
>
> There may be some misunderstanding here. ``Different concept libraries are not concatenated but utilized separately``.
>
> According to L134-138, we conduct experiments with different concept libraries separately to ensure the general applicability of our conclusions. Concepts from different libraries are not concatenated during the experiment.
>
> For example, all types of concepts are utilized separately in Figure 3&6. Only textual concepts are utilized in Figure 4&7 and Table 4&5.
>
> **Revision**. In the revision, we have illustrated the separate usage of different concept libraries in L137-139.

---

> ### Author Response · Authors · 2024-11-20
> **Official Comment by Authors (2/2)**
>
> ## Q4: Comparisons with DEAL
>
> Thank you for mentioning this interesting work that further supports our findings.
>
> **Performance comparisons**. Following your suggestion, the performance comparisons on ImageNet are provided as follows.
>
> | Model | CLIP-RN50 | CLIP-ViT-B/32 |
> | ----- | --------- | ------------- |
> | DEAL  | 70.0      | 70.8          |
> | Ours  | **70.8**  | **73.0**      |
>
> Within the same pre-trained model, our method achieves better performance than DEAL.
>
> Then, we discuss the `differences` between our method and DEAL.
>
> **Training strategy**.
>
> - DEAL is specifically designed for the CLIP architecture and employs contrastive learning methods to fine-tune the entire model.
> - Our method can be applied to arbitrary architectures and only fine-tune the visual encoder.
>
> **Prediction mechanism**.
> - DEAL predicts based on the average embedding similarity between the image and concepts within the category.
> - Our method inputs the image representations to a prediction head without the text encoder.
>
> **Interpretability-classifiability relationship**.
> - DEAL solely observed a `qualitative` impact of interpretability on classifiability in `CLIP`.
> - In comparison, we find the `mutually promoting` relationship between both factors on a `wide range of models` in a `quantitative` manner.
>
> **Revision**. In the revision, we have added the discussion of differences between our methods and DEAL in Section A.5.4.
>
>
> [1] Liu, Z., et al. (2022). A convnet for the 2020s. CVPR.
>
> [2] Liu, Z., et al. (2021). Swin transformer: Hierarchical vision transformer using shifted windows. ICCV.
>
> [3] Oikarinen, T., et al. (2023). Label-free concept bottleneck models. ICLR.

---

> ### Comment · Reviewer_d5fZ · 2024-11-21
>
> Thank you for your detailed responses to my questions. I have a few additional inquiries:
>
>
> Q1. Could you specify the dataset used in Table A14?
>
> Q2. In Section A.1.1, was the description for the textual concept included in the original experimental setup? If not, could you clarify any changes observed in the experiment and provide an explanation of them?

---

> > ### Author Response · Authors · 2024-11-22
> > **Official Comment by Authors**
> >
> > ## Q1: Dataset used in Table A14
> > Thank you for your careful review! All models are evaluated on the ADE20K dataset [1], a standard benchmark for segmentation tasks.
> >
> > **Revision**. We have added the name of the utilized dataset in the caption of Table A14.
> >
> > ## Q2: Extraction of textual concepts
> > ``The description for the textual concept in A.1.1 has been included in the original setup for all experiments.`` In the original manuscript, the details of this part were inadvertently omitted, so we sincerely appreciate your efforts to enhance our presentation of the experimental details!
> >
> > [1] Zhou, B., et al. (2017). Scene parsing through ade20k dataset. CVPR (pp. 633-641).

---

> > > ### Comment · Reviewer_d5fZ · 2024-11-22
> > >
> > > Thank you for addressing all my questions. While my concerns about the novelty of this work remain, as papers [1,2,3] have also explored the positive correlation between interpretability and classifiability, the main contribution here is sent to the design of IIS. Based on this, I have raised my score to 6.
> > >
> > > [1] Zhao, Chenyang, et al. (2024) Gradient-based Visual Explanation for Transformer-based CLIP. International Conference on Machine Learning. PMLR.
> > >
> > > [2] Li, T., Ma, M., & Peng, X. (2025). Deal: Disentangle and localize concept-level explanations for vlms. In European Conference on Computer Vision (pp. 383-401). Springer, Cham.
> > >
> > > [3] Gandelsman, Y., Efros, A. A., & Steinhardt, J. (2023). Interpreting CLIP's Image Representation via Text-Based Decomposition. arXiv preprint arXiv:2310.05916.

---

> > > > ### Author Response · Authors · 2024-11-22
> > > >
> > > > Thank you very much for your supportive feedback! Compared with methods [1,2,3], our method may have similar motivations in exploring the relationship between interpretability and classifiability. However, our contributions are different from them.
> > > >
> > > > **Methods**. In contrast to these papers that focus on designing interpretation methods, we propose ``a general metric`` of representation interpretability to ``quantitatively investigate`` the interpretability-classifiability relationship.
> > > >
> > > > **Investigation Results**. While previous approaches have observed that enhancing interpretability improves classifiability (interpretability$\rightarrow$classifiability), we have discovered ``a mutually promoting relationship`` (interpretability$\leftrightarrow$classifiability).
> > > >
> > > > Inspired by your suggestions, we will further incorporate interpretability enhancements from these methods [1,2,3] into our quantitative measures in future work.
> > > >
> > > > **Revision**. We have updated the revision by incorporating the discussion with these methods [1,2,3] in Section A.5.4.
> > > >
> > > > [1] Zhao, Chenyang, et al. (2024) Gradient-based Visual Explanation for Transformer-based CLIP. ICML.
> > > >
> > > > [2] Li, T., Ma, M., & Peng, X. (2025). Deal: Disentangle and localize concept-level explanations for vlms. ECCV.
> > > >
> > > > [3] Gandelsman, Y., Efros, A. A., & Steinhardt, J. (2023). Interpreting CLIP's Image Representation via Text-Based Decomposition. arXiv.

---

### Official Review · Reviewer_AruE · 2024-11-04

**Soundness:** 4
**Presentation:** 4
**Contribution:** 3
**Rating:** 8
**Confidence:** 4

**Summary:**

This paper proposes the IIS metric to measure the interpretability of representations from image encoders and demonstrates a positive correlation between the interpretability and classifiability of the classifiability-based pre-trained representations.

**Strengths:**

1. This paper quantifies the interpretability of representations and discovers an interesting and significant positive correlation between the interpretability and classifiability of representations. The results have been demonstrated on multiple datasets and architectures.
2. The observation is used in designing IIS maximization loss, which results in improved classifiability.
3. The paper is well-written, making it easy to follow the main ideas and claims.

**Weaknesses:**

1. The proposed IIS score is similar to the “average accuracy obtained at different number of effective concepts” proposed in [1]. Particularly, [1] trains the final linear classifier with elastic-net regularization in GLM-SAGA and tunes regularization strength for controlling sparsity (number of effective concepts). The average accuracy is obtained by averaging accuracy at different sparsity levels. The similarities between the metric proposed in [1] and IIS should be discussed.
2. Limited dataset results: To demonstrate the applicability of this method in wide scenarios, the results should be provided on additional datasets, including Actions: UCF-101, Textures: DTD, and Skin tumors: HAM10000 following [2].
3. Table 5 lacks comprehensive comparisons to methods [3, 4]. Particularly, [3] can work for any backbone network architecture, including ViT and Swin.

[1] Srivastava, Divyansh, Ge Yan, and Tsui-Wei Weng. VLG-CBM: Training Concept Bottleneck Models with Vision-Language Guidance. arXiv preprint arXiv:2408.01432 (2024).

[2] Yue Yang, Artemis Panagopoulou, Shenghao Zhou, Daniel Jin, Chris Callison-Burch, and Mark Yatskar. Language in a bottle: Language model guided concept bottlenecks for interpretable image classification. In CVPR, 2023.

[3] Tuomas P. Oikarinen, Subhro Das, Lam M. Nguyen, and Tsui-Wei Weng. Label-free concept bottleneck models. In The Eleventh International Conference on Learning Representations, ICLR 2023

[4] An Yan, Yu Wang, Yiwu Zhong, Chengyu Dong, Zexue He, Yujie Lu, William Yang Wang, Jingbo Shang, and Julian McAuley. Learning concise and descriptive attributes for visual recognition. In ICCV, 2023.

**Questions:**

Please see weaknesses

---

> ### Author Response · Authors · 2024-11-20
>
> We would like to thank you for the valuable comments! We will diligently follow your guidance to further enrich our experiments.
>
> ## W1: Comparisons between IIS and AVG Acc. in VLG-CBM
>
> Thank you for introducing this work to us.
> The main differences between IIS and VLG-CBM can be summarized in three aspects:
>
> **Metric Definition**.
> - The IIS is defined as the area under the curve formed by the sparsity and accuracy.
> - The metric in VLG-CBM is solely defined based on the average accuracy at different sparsity levels.
>
> **Objective**.
>
> - The objective of IIS is to quantify the interpretability of arbitrary vision pre-trained representations.
>
> - The average accuracy in VLG-CBM serves as a metric for evaluating the performance of CBMs, a specific type of interpretable model.
>
> **Sparsification**.
>
> - Our sparsification mechanism can accurately control the number of concepts for prediction during forward propagation.
>
> - VLG-CBM adjusts the number of concepts by the sparsity regularization in the loss function, which requires complex strategies including gradually reducing the regularization strengths during training and post-training weight pruning.
>
> **Revision**: We have incorporated the discussion of IIS and the metric in VLG-CBM in Section A.5.3 of the revision.
>
> ## W2: Experiments on additional datasets
>
> Following your suggestion, we conduct experiments on UCF-101, DTD and HAM10000.
>
> **Interpretability-classifiability relationship**. The investigation results in our manuscript, i.e., the positive correlation between interpretability and classifiability, can be generalized to these three datasets.
>
> | Model      | UCF101-IIS | DTD-IIS | HAM10000-IIS |
> | --- | --- | --- | ---- |
> | ResNet-18  | 0.949 | 0.934   | 0.912  |
> | ResNet-50  | 0.967 | 0.940   | 0.914  |
> | ResNet-152 | 0.976 | 0.957   | 0.915  |
>
> **Performance comparisons**. We provide comparisons with LaBo on these three datasets. Our method achieves better performance than LaBo.
>
> | Model | UCF101 | DTD   | HAM10000 |
> | ----- | ------ | ----- | -------- |
> | LaBo  | 97.68  | 76.86 | 81.40    |
> | Ours  | **97.79**  | **77.22** | **81.90**    |
>
> **Revision**: We have incorporated the additional experiments in Section A.2.3 of the revision.
>
> ## W3: More comparisons with existing methods
> Thank you for your suggestions that enrich our experiments.
>
> **Experiments**. Following your suggestion, we add comparisons with LF-CBM [1] on CUB-200 and ImageNet-1k datasets based on different backbones.
>
> | Model          | ImageNet | CUB-200 |
> | -------------- | -------- | ------- |
> | LF-CBM (ViT-B) |  77.94   | 68.97   |
> | Ours (ViT-B)   |  **78.50**   | **69.14**   |
> | LF-CBM (ViT-L) |  85.52   | 82.83   |
> | Ours (ViT-L)   |  **86.85**   | **83.51**   |
>
> **Analysis**. Our method exhibits better performance than LF-CBM. Moreover, the performance of LF-CBM is significantly improved by leveraging representations with higher classifiability. This phenomenon aligns with the discussion in Section A.5.2.
>
> For [2], its performance has only been reported in few-shot scenarios, which deviates from our experimental settings. Nevertheless, the impact of training data volumes on the interpretability-classifiability relationship is an interesting question for investigation in future work.
>
> **Revision**: We have incorporated the additional experiments in Section A.2.3 and [2] in Section A.5.2 of the revision.
>
> [1] Oikarinen, T., et al. (2023). Label-free concept bottleneck models. ICLR.
>
> [2] Yan, A., et al. (2023). Learning concise and descriptive attributes for visual recognition. ICCV.

---

> > ### Comment · Reviewer_AruE · 2024-11-24
> >
> > I thank authors for their comprehensive experiments and clarifying my concerns. I have raised my score to 8 since all my concerns have been answered.

---

> > > ### Author Response · Authors · 2024-11-24
> > >
> > > Dear reviewer, we sincerely appreciate your positive feedback and support. Your suggestions have helped us improve our work. We are glad that our responses addressed your concerns.

---

### Official Review · Reviewer_Hosp · 2024-11-05

**Soundness:** 4
**Presentation:** 4
**Contribution:** 4
**Rating:** 8
**Confidence:** 5

**Summary:**

This paper addresses the critical question of whether high interpretability and classifiability can be achieved simultaneously in pre-trained vision models. To this end, the authors propose the Inherent Interpretability Score (IIS), a novel metric designed to quantitatively evaluate the interpretability of pre-trained representations. Through extensive experimentation, the authors find a positive correlation between interpretability and classifiability. Moreover, they observe that maximizing interpretability can further enhance model performance, which, in turn, improves the performance of interpretability-based models.

**Strengths:**

* This paper utilizes concept ablation method for evaluation the interpretability of the representations, which has also been adopted for evaluating the post-hoc interpretation methods [1].
* The authors utilize both visual and textual concepts for IIS.
* The paper is well-written and easy to follow.
* The authors provide code for reproducibility check.

**Weaknesses:**

1. **Completeness of Evaluation:** Huang et al. [2] have discussed the trustworthiness in concept bottleneck models (CBMs), noting that several concepts can be activated in irrelevant regions with regards to the input image, leading to a loss of model interpretability. However, the authors have overlooked this issue.
2. **Method:** In the IIS evaluation, in what order are concepts removed? How would the evaluation results differ if removing began with the most important concepts for the category versus the least important?
3. **Experiments:** If the visual model’s features have already been aligned with language, as in CLIP [3], would the interpretability be significantly higher compared to a standard ViT?

> [1] Hooker, Sara, et al. "A benchmark for interpretability methods in deep neural networks." Advances in neural information processing systems 32 (2019).
>
> [2] Huang, Qihan, et al. "On the Concept Trustworthiness in Concept Bottleneck Models." Proceedings of the AAAI Conference on Artificial Intelligence. Vol. 38. No. 19. 2024.
>
> [3] Radford, Alec, et al. "Learning transferable visual models from natural language supervision." International conference on machine learning. PMLR, 2021.

**Questions:**

My questions are listed in the Weakness section.

---

> ### Author Response · Authors · 2024-11-20
>
> Thank you very much for your valuable comments, which help us better clarify our contributions and inspire our future work! We hope the following responses can address your concerns.
>
> ## W1: Completeness of evaluation
>
> Your concern is very insightful and prompts us to discuss the trustworthiness of concept-based interpretations.
>
> **Directly incorporating trustworthiness limits the usability of IIS**. We agree with you that similar to the sparsity, trustworthiness serves as another important factor for evaluating the interpretation quality. However, existing evaluations of trustworthiness heavily rely on ``concept annotations`` and cannot be applied to arbitrary datasets.
> Therefore, we did not incorporate this factor into IIS as it would confine the application scope of IIS solely to datasets with concept annotations, which contradicts our objective of applying IIS to arbitrary datasets.
>
> **Future plans for incorporating trustworthiness**. Nonetheless, we recognize the crucial role of trustworthiness in concept-based interpretations. In future work, we will try to evaluate the trustworthiness automatically by incorporating an auxiliary visual language grounding model, and incorporate this factor into the measurement of representation interpretability.
>
> **Revision**: In the revision, we have added the discussion on future work of refining IIS by incorporating trustworthiness in Section 6.
>
> ## W2: Removing orders of concepts
>
> **Ascending in the manuscript**. As Equation 4 shows, the concepts are removed in ``ascending`` order based on their similarity to the image representations. Concepts with the least similarity to image semantics are removed first.
>
> **Descending order**. Following your suggestion, we report the IIS of different representations by removing concepts in ``descending`` order based on their similarity to the image representations.
>
> | Model      | ResNet-18 | ResNet-34 | ResNet-50 | ResNet-101 | ResNet-152 |
> | ---------- | --------- | --------- | --------- | ---------- | ---------- |
> | Ascending  | 0.695     | 0.719     | 0.766     | 0.776      | 0.780      |
> | Descending | 0.284     | 0.298     | 0.340     | 0.352      | 0.354      |
>
> **Analysis**. When reversing the order of concept removal, IIS demonstrates a significant decrease due to the projection of representations onto less relevant concepts for predictions.
>
> However, representations with higher classifiability still demonstrate higher IIS under the descending sparsification order. This phenomenon further demonstrates that the positive correlation between interpretability and classifiability is an `inherent characteristic` of pre-trained representations, rather than a consequence of specific sparsification methods.
>
> **Revision**. In the revision, we have incorporated the discussions of concept removal orders in Section A.2.1.
>
> ## W3: Experiments with CLIP
>
> **Experiments**. Following your suggestion, we provide the comparison results between ViT-B/16 and CLIP-ViT-B/16 on the CUB-200 dataset.
>
> | Model          | IIS  | Acc@1 |
> | -------------  |---- | ----- |
> | ViT-B/16       |0.764| 75.9  |
> | CLIP-ViT-B/16  |0.808| 80.6  |
>
> **Analysis**. The experimental results remain consistent with our investigations in Section 3.2, i.e., representations with higher classifiability exhibit enhanced interpretability. Representations pre-trained with large-scale vision-language datasets have stronger generalization ability than representations of the standard ViT, thereby resulting in improved classifiability on this dataset and subsequently enhancing their interpretability.
>
> **Revision**. We have added these experiments in Section A.2.3 of the revision.

---

> ### Comment · Reviewer_Hosp · 2024-11-21
>
> Thanks for the authors' rebuttal. For the evaluation of trustworthiness, could the authors consider using visualization approach for qualitative validation? For instance, visualizing the activation locations of each concept and comparing them with the input images.

---

> > ### Author Response · Authors · 2024-11-22
> >
> > Thank you for your constructive suggestions that enable verifying the trustworthiness of interpretable predictions based on powerful pre-trained representations!
> >
> > Following your suggestion, we visualize the activation locations of each concept through Grad-CAM and compare them with the input images. As the figure in (https://anonymous.4open.science/r/FigureA11_of_IIS-60B8/trust_all.png) shows, the activation regions for each concept are related to the concept semantic, thereby qualitatively supporting the trustworthiness of leveraging powerful pre-trained representations~(e.g., ViT-L) for interpretable predictions.
> >
> > **Revision**. We have incorporated these visualizations in Section A.2.5 and Figure A11.

---

> > > ### Comment · Reviewer_Hosp · 2024-11-22
> > >
> > > Thanks for the authors' reply. All my concerns have been addressed. Therefore, I raise my score to 8.

---

> > > > ### Author Response · Authors · 2024-11-22
> > > >
> > > > We sincerely appreciate your feedback and are glad that our responses addressed your questions. We will further explore the quantification of representation interpretability following your guidance in future work.

---

### Official Review · Reviewer_xfXM · 2024-11-08

**Soundness:** 3
**Presentation:** 3
**Contribution:** 3
**Rating:** 8
**Confidence:** 4

**Summary:**

This paper proposes the Inherent Interpretability Score (IIS) as a novel quantitative metric for evaluating the interpretability of the pre-trained representation. The authors demonstrate the positive correlation between the interpretability and classifiability of the classifiability-oriented pre-trained representations through extensive experiments.

**Strengths:**

1. The authors proposed a novel evaluation metric to quantitatively evaluate the pre-trained representation's interpretability. The proposed IIS can not only be used in the research area related to post-hoc explainable approaches but also improve model classification performance and interoperability.
2. The paper is written and well-organized. It is easy to follow the authors' ideas and understand their approaches. The authors clearly show their motivation and ideas in Figure 1 and Figure 2. The notations and experimental results are clear and easy to read.
3. The authors have conducted extensive experiments across various datasets with different pre-trained models to show the effectiveness of IIS. The experimental results support the authors' claims from several different aspects, for example, the positive correlation, factor analysis,  and applications of IIS.

**Weaknesses:**

1. Besides extensive experimental results, the authors should have provided clear theoretical analyses of the proposed method.
2. The proposed method IIS relies on pre-defined or generated concepts, which are usually hard to get and not applicable in real-world scenarios.

**Questions:**

1. How do you deal with the linear weights in the additional linear classifier, as shown in Eq. (5)?

---

> ### Author Response · Authors · 2024-11-20
>
> We sincerely thank you for your valuable feedback on our manuscript. Your comments have been instrumental in guiding our future work.
>
> ## W1: Theoretical analyses of the proposed method
>
> Thank you for your valuable suggestions! We agree with you that adding theoretical analysis can further enhance the generality of our experimental results.
>
> Inspired by the concept disentangle phenomenon observed in [1], we will take the representation disentangle theory [2] in pre-trained representations as a starting point. Then, we will theoretically analyze whether the vectors corresponding to concepts exhibit enhanced disentanglement, thereby enabling descriptions of pre-trained representations with less information loss in a sparse manner.
>
> **Revision**. In the revision, we have added the discussion of future work on theoretical analysis for IIS in Section 6.
>
> ## W2: Applicability in real-world scenarios
> Thank you for your constructive comments! We want to clarify that our concept libraries are not difficult to acquire and can be applied to real-world scenarios.
>
>
> **Automated concept acquisition**. The construction of concept libraries can be accomplished automatically without human annotations.
> The visual concepts are constructed by aggregating image patches and segments, and the textual concepts are generated by prompting an LLM and aligned with images using a vision-language model.
>
> **Experiments on real-world datasets**. We have conducted experiments on real-world datasets without concept annotations, including ImageNet-1k (Figures 3&5&6&7&9, Table 5) and Kinetics-400 (Figure A12).
>
>
> ## Q1: Weights in the additional linear classifier
> The classifier can be used in two cases in Section 4.1 and 4.2.
>
> In Section 4.1, the output of this classifier is used to compute the auxiliary loss in Equation 8.
>
> In Section 4.2, the classifier provides interpretable predictions based on concepts.
>
> **Revision**. We have mentioned the usage of this classifier in Section 4.2 of the revision.
>
> [1] Materzyńska, J., Torralba, A., & Bau, D. (2022). Disentangling visual and written concepts in CLIP. CVPR.
>
> [2] Wang, X., Chen, H., Wu, Z., & Zhu, W. (2024). Disentangled representation learning. IEEE TPAMI.

---

> > ### Comment · Reviewer_xfXM · 2024-11-21
> > **Thank you for your responses!**
> >
> > Thank you for your responses! My concerns have been addressed. I will keep my score of 8, accept, good paper.

---

> > > ### Author Response · Authors · 2024-11-22
> > >
> > > Dear Reviewer, we sincerely appreciate your support and recognition. We are committed to diligently incorporating your suggestions into our future work!

---

### Author Response · Authors · 2024-11-20

We would like to thank all the reviewers for their diligent efforts and valuable suggestions, which have greatly contributed to improving the quality of our manuscript.

**Summary of strengths**:

We sincerely appreciate that you find our method:

- novel and interesting (reviewers xfXM, AruE, d5fZ, wUp3, and 7e64);

- provide extensive evaluations and convincing results (reviewers xfXM, Hosp, AruE, d5fZ, 511b, and wUp3);

- show broad application direction (reviewers xfXM, d5fZ, and 511b);

- well-organized and well-written (reviewers xfXM, Hosp, and AruE);

- provide a quantitive evaluation for interpretability (reviewers xfXM, Hosp, AruE, d5fZ, 511b, and wUp3);

- help improve classifiability (reviewers xfXM, AruE, and 511b).

---

### Meta-Review · Area_Chair_JrXg · 2024-12-04

**Metareview:**

This paper studies the tradeoffs between interpretability and classifiability, finding empirically a strong correlation. For this, they propose a new evaluation metric that is usable for both explainability and improving classification performance. The reviewers unanimously agree that this is a good paper and should be accepted. The main weakness is the limited theoretical explanation for this phenomenon, but all the reviewers agree that the experimental results are convincing.

**Additional Comments On Reviewer Discussion:**

There was extensive discussion with the reviewers, especially around new experiments (e.g., more baselines, segmentation, more data sets). Overall, the reviewers were satisfied with the improvements and raised their scores.

---

### Decision · Program_Chairs · 2025-01-22

Accept (Spotlight)